# Time-uniform confidence bands for the CDF under nonstationarity

**Paul Mineiro**
Microsoft Research
pmineiro@microsoft.com

**Steve Howard**
The Voleon Group
steve@stevehoward.org

## Abstract

Estimation of a complete univariate distribution from a sequence of observations is a useful primitive for both manual and automated decision making. This problem has received extensive attention in the i.i.d. setting, but the arbitrary data dependent setting remains largely unaddressed. We present computationally felicitous time-uniform and value-uniform bounds on the CDF of the running averaged conditional distribution of a sequence of real-valued random variables. Consistent with known impossibility results, our CDF bounds are always valid but sometimes trivial when the instance is too hard, and we give an instance-dependent convergence guarantee. The importance-weighted extension is appropriate for estimating complete counterfactual distributions of rewards given data from a randomized experiment, e.g., from an A/B test or a contextual bandit.

## 1 Introduction

What would have happened if I had acted differently? Although as old as time itself, successful companies have recently embraced this question via offline estimation of counterfactual outcomes using data from existing randomized experiments or contextual bandits. The problem is important in diverse domains such as software testing [Lindon et al., 2022, Wang and Chapman, 2022], portfolio management [Liu, 2021], and medicine [Shen et al., 2022]. These experiments are run in the real (digital) world, which is rich enough to demand non-asymptotic statistical techniques under non-parametric and non-stationary (i.e., not i.i.d.) models. Although existing methods apply for estimating *average* outcomes in this general setting (either under the observed distribution or counterfactual ones), estimating a complete *distribution* of outcomes is heretofore only possible with additional assumptions: see Table 1 for a summary and Section 5 for complete discussion of related work.

To fix ideas, we briefly describe an application from Lindon et al. [2022] in the context of canary testing: rolling out changes in an online service to a small, random subset of users in order to detect accidental performance regressions while minimizing effect on overall user experience. The metric of interest measures latency for fetching content from the service. It is common to look beyond the mean of the latency distribution and especially to check for regressions in upper quantiles. As such, the authors choose to estimate bounds on the entire CDF of this latency metric under both the control and treatment arms and check for a statistically significant differences at any point in the CDF. The hope is to detect regressions as soon as possible, often within seconds or minutes, so the authors employ a sequential method which allows an automated system to continuously update the CDF bounds as data accumulates and to stop as soon as a significant regression is detected. Statistically, this translates into the requirement of confidence bands for the CDF which are both uniform over time (valid after every update) and uniform over values (so we can check for regressions at any quantile). We seek such bounds whose statistical validity is guaranteed under a minimum of assumptions.

Intriguingly, this problem is provably impossible in the general data dependent setting [Rakhlin et al., 2015]. Consequently, our bounds always achieve non-asymptotic coverage, but may converge to zero width slowly or not at all, depending on the hardness of the instance. We call this design principle AVAST (Always Valid And Sometimes Trivial).

37th Conference on Neural Information Processing Systems (NeurIPS 2023).

Table 1: Comparison to prior art for quantile-uniform CDF estimation. *Time-uniform*: a sequence of confidence bands whose coverage holds uniformly over time with high probability. *Nonstationary*: does not require an i.i.d. assumption. *Non-asymptotic*: guarantees hold at all sample sizes. *Non-parametric*: guarantees apply over an infinite-dimensional class of distributions. *Counterfactual*: method applies to estimating a distribution other than the one which generated the observed data, via importance weighting. $w_{\max}$-*free*: guarantee applies without a bound on the maximum importance weight. See Section 5 for details.

| REFERENCE | TIME-UNIFORM? | NON-STATIONARY? | NON-ASYMPTOTIC? | NON-PARAMETRIC? | COUNTER-FACTUAL? | $w_{\max}$-FREE? |
|---|---|---|---|---|---|---|
| HR22 | ✓ | | ✓ | ✓ | | N/A |
| HLLA21 | | | ✓ | ✓ | ✓ | |
| UNO21, [IID] | | | ✓ | ✓ | ✓ | ✓ |
| UNO21, [NS] | | ✓ | | | ✓ | ✓ |
| WS22, [§4] | ✓ | | ✓ | ✓ | ✓ | ✓ |
| THIS PAPER | ✓ | ✓ | ✓ | ✓ | ✓ | ✓ |

**Contributions**

1. In Section 3.2 we provide a time- and value-uniform upper bound on the CDF of the averaged historical conditional distribution of a discrete-time real-valued random process. Consistent with the lack of sequential uniform convergence of linear threshold functions [Rakhlin et al., 2015], the bounds are Always Valid (see Theorem 3.1) And Sometimes Trivial, i.e., the width guarantee is instance-dependent (see Theorem 3.3) and may not converge to zero width in the infinite data limit. When the data generating process is smooth with respect to the uniform distribution on the unit interval, the bound width adapts to the unknown smoothness parameter, following the framework of smoothed online learning [Rakhlin et al., 2011, Haghtalab et al., 2020, 2022b,a, Block et al., 2022].

2. In Section 3.3 we extend the previous technique to distributions with support over the entire real line, and further to distributions with a known countably infinite or unknown nowhere dense set of discrete jumps; with analogous instance-dependent guarantees.

3. In Section 3.4 we extend the previous techniques to importance-weighted random variables, achieving our ultimate goal of estimating a complete counterfactual distribution of outcomes.

We exhibit our techniques in various simulations in Section 4. Computationally our procedures have comparable cost to point estimation of the empirical CDF, as the empirical CDF is a sufficient statistic.

## 2    Problem Setting

Let $(\Omega, \mathcal{F}, \{\mathcal{F}_t\}_{t\in\mathbb{N}}, \mathbb{P})$ be a probability space equipped with a discrete-time filtration, on which let $X_t$ be an adapted, real-valued random process. Let $\mathbb{E}_t[\cdot] \doteq \mathbb{E}[\cdot|\mathcal{F}_t]$. The quantity of interest is the (random) map $\overline{\text{CDF}}_t : \mathbb{R} \to [0, 1]$, the CDF of the averaged historical conditional distribution at time $t$:

$$\overline{\text{CDF}}_t(v) \doteq \frac{1}{t} \sum_{s \leqslant t} \mathbb{E}_{s-1}\left[1_{X_s \leqslant v}\right]. \tag{1}$$

We desire simultaneously time- and value-uniform bounds which hold with high probability, i.e., adapted sequences of maps $L_t, U_t : \mathbb{R} \to [0, 1]$ satisfying

$$\mathbb{P}\left(\begin{smallmatrix}\forall t\in\mathbb{N}\\\forall v\in\mathbb{R}\end{smallmatrix} : L_t(v) \leqslant \overline{\text{CDF}}_t(v) \leqslant U_t(v)\right) \geqslant 1 - 2\alpha. \tag{2}$$

Note that the estimand $\overline{\text{CDF}}_t$ is changing at each time step as we incorporate the conditional distribution of the latest observation into our historical average. The maps $L_t, U_t$ provide a sequence of confidence bands which contain this sequence of changing CDFs uniformly over time with high probability.

In the i.i.d. setting, Equation (1) is deterministic and independent of $t$, reducing to the CDF of the (unknown) generating distribution. In this setting, the classic results of Glivenko [1933] and

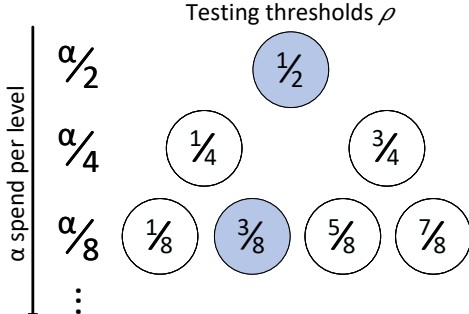

Testing thresholds $\rho$

Figure 1: Visualization of Algorithm 1. The values of interest are uncountably infinite; the algorithm allocates probability to maintain upper bounds on a countably infinite set of points $\rho$ at different resolution levels via the monotonicity of $\overline{\text{CDF}}_t(v)$. As resolution increases, the value $\rho$ better approximates $v$, but the allocated probability decreases; the algorithm chooses the tightest of available bounds. Shaded nodes would be consulted for an upper bound for $v = 5/17$.

---

**Algorithm 1** Unit Interval Upper Bound. $\epsilon(d)$ is an increasing function specifying the resolution of discretization at level $d$. $\Xi_t\left(\rho; \delta, d, \Psi_t\right)$ is an upper confidence sequence for fixed value $\rho$ with coverage at least $(1 - \delta)$.

---

**Input:** value $v$; confidence $\alpha$; sufficient statistic $\Psi_t$.
// **e.g.** $\Psi_t \doteq X_{1:t}$ **or** $\Psi_t \doteq (W_{1:t}, X_{1:t})$
**Output:** $U_t(v)$ satisfying Equation (2).
**if** $v > 1$ **then return** 1 **end if**
$u \leftarrow 1$
$v \leftarrow \max(0, v)$
**for** $d = 1$ **to** $\infty$ **do**
   $\rho_d \leftarrow \epsilon(d)^{-1}\lceil \epsilon(d)v \rceil$
   $\delta_d \leftarrow \alpha/(d(d+1)\epsilon(d))$
   $u \leftarrow \min(u, \Xi_t(\rho_d; \delta_d, \Psi_t))$
   **if** $0 = \sum_{s \leqslant t} 1_{X_s \in (v, \rho_d]}$ **then**
      **return** $u$
   **end if**
**end for**

---

Cantelli [1933] established uniform convergence of linear threshold functions; subsequently the Dvoretzky-Kiefer-Wolfowitz (DKW) inequality characterized fixed-time and value-uniform convergence rates [Dvoretzky et al., 1956, Massart, 1990]; extended later to simultaneously time- and value-uniform bounds [Howard and Ramdas, 2022]. The latter result guarantees an $O(t^{-1}\log(\log(t)))$ confidence interval width, matching the limit imposed by the Law of the Iterated Logarithm.

**AVAST principle** In contrast, under arbitrary data dependence, linear threshold functions are not sequentially uniformly convergent, i.e., the averaged historical empirical CDF does not necessarily converge uniformly to the CDF of the averaged historical conditional distribution [Rakhlin et al., 2015]. Consequently, additional assumptions are required to provide a guarantee that the confidence width decays to zero. In this paper we design bounds that are Always Valid And Sometimes Trivial, i.e., under worst-case data generation, $\sup_v |U_t(v) - L_t(v)| = O(1)$ as $t \to \infty$. Fortunately our bounds are also equipped with an instance-dependent width guarantee based upon the smoothness of the distribution to a reference measure qua Definition 3.2.

**Additional Notation** Let $X_{a:b} = \{X_s\}_{s=a}^b$ denote a contiguous subsequence of a random process. Let $\mathbb{P}_t$ denote the average historical conditional distribution, defined as a (random) distribution over the sample space $\mathbb{R}$ by $\mathbb{P}_t(A) \doteq t^{-1}\sum_{s \leqslant t} \mathbb{E}_{s-1}\left[1_{X_s \in A}\right]$ for a Borel subset $A$ (note $\mathbb{P}_t$ represents the entire historical average while $\mathbb{E}_t$ corresponds to a single conditional distribution).

## 3 Derivations

### 3.1 High Level Design

Our approaches work as reductions, achieving the value- and time-uniform guarantee of Equation (2) by combining bounds $\Lambda_t, \Xi_t$ that satisfy a time-uniform guarantee at any fixed value $\rho$,

$$\mathbb{P}\left(\forall t \in \mathbb{N} : \Lambda_t(\rho) \leqslant \overline{\text{CDF}}_t(\rho) \leqslant \Xi_t(\rho)\right) \geqslant 1 - \delta(\rho). \tag{3}$$

The bounds $\Lambda_t, \Xi_t$ are tools for estimating a sequence of *scalars*, in this case $(\overline{\text{CDF}}_t(\rho))_{t=1}^\infty$ for a fixed value $\rho$. We show how to extend such tools to the more difficult problem of estimating a sequence of (cumulative distribution) *functions*.

There are multiple existing approaches to obtaining the guarantee of Equation (3): we provide a self-contained introduction in Appendix A. For ease of exposition, we will only discuss how to construct

a time- and value-uniform upper bound by combining fixed-value, time-uniform upper bounds, and defer the analogous lower bound construction to Appendix B.2. Our approach is to compose these fixed-value bounds into a value-uniform bound by taking a union bound over a particular collection of values, leveraging monotonicity of the CDF.

**Quantile vs Value Space**   In the i.i.d. setting, a value-uniform guarantee can be obtained by taking a careful union bound over the unique value associated with each quantile [Howard and Ramdas, 2022]. This "quantile space" approach has advantages, e.g., variance based discretization and covariance to monotonic transformations. However, under arbitrary data dependence, the value associated with each quantile can change. Therefore we proceed in "value space". See Appendix A.1 for more details.

### 3.2   On the Unit Interval

Algorithm 1, visualized in Figure 1, constructs an upper bound on Equation (1) which, while valid for all values, is designed for random variables ranging over the unit interval. For a given value $v$, it searches over upper bounds on the CDF evaluated at a decreasing sequence of values $\rho_1 \geqslant \rho_2 \geqslant \cdots \geqslant v$ and exploits monotonicity of $\overline{\mathrm{CDF}}_t(v)$. That is, at each level $d = 1, 2, \ldots$, we construct a discretizing grid of size $\epsilon(d)$ over the unit interval, and construct a time-uniform upper bound on $\overline{\mathrm{CDF}}_t(\rho)$ for each grid point $\rho$ using the fixed-value confidence sequence oracle $\Xi_t$. Then, for a given value $v$, at each level $d$ we make use of the fixed-value confidence sequence for smallest grid point $\rho_d \geqslant v$, and we search for the level $d$ which yields the minimal upper confidence bound. A union bound over the (countably infinite) possible choices for $\rho_d$ controls the coverage of the overall procedure. Because the error probability $\delta_d$ decreases with $d$ (and the fixed-value confidence radius $\Xi_t$ increases as $\delta$ decreases), the procedure can terminate whenever no observations remain between the desired value $v$ and the current upper bound $\rho_d$, as all subsequent bounds are dominated.

The lower bound is derived analogously in Algorithm 2 (which we have left to Appendix B.2 for the sake of brevity) and leverages a lower confidence sequence $\Lambda_t(\rho; \delta, \Psi_t)$ (instead of an upper confidence sequence) evaluated at an increasingly refined lower bound on the value $\rho \leftarrow \epsilon(d)^{-1}\lfloor \epsilon(d)v \rfloor$.

**Theorem 3.1.** *If $\epsilon(d) \uparrow \infty$ as $d \uparrow \infty$, then Algorithms 1 and 2 terminate with probability one. Furthermore, if for all $\rho$, $\delta$, and $d$ the algorithms $\Lambda_t(\rho; \delta, \Psi_t)$ and $\Xi_t(\rho; \delta, \Psi_t)$ satisfy*

$$P(\forall t : \overline{\mathrm{CDF}}_t(\rho) \geqslant \Lambda_t(\rho; \delta, \Psi_t)) \geqslant 1 - \delta, \tag{4}$$

$$P(\forall t : \overline{\mathrm{CDF}}_t(\rho) \leqslant \Xi_t(\rho; \delta, \Psi_t)) \geqslant 1 - \delta, \tag{5}$$

*then guarantee (2) holds with $U_t, L_t$ given by the outputs of Algorithms 1 and 2, respectively.*

*Proof.* See Appendix B.3. □

Theorem 3.1 ensures Algorithms 1 and 2 yield the desired time- and value-uniform coverage, essentially due to the union bound and the coverage guarantees of the oracles $\Xi_t, \Lambda_t$. However, coverage is also guaranteed by the trivial bounds $0 \leqslant \overline{\mathrm{CDF}}_t(v) \leqslant 1$. The critical question is: what is the bound width?

**Smoothed Regret Guarantee**   Even assuming $X$ is entirely supported on the unit interval, on what distributions will Algorithm 1 provide a non-trivial bound? Because each $[\Lambda_t(\rho; \delta, \Psi_t), \Xi_t(\rho; \delta, \Psi_t)]$ is a confidence sequence for the mean of the bounded random variable $1_{X_s \leqslant \rho}$, we enjoy width guarantees at each of the (countably infinite) $\rho$ which are covered by the union bound, but the guarantee degrades as the depth $d$ increases. If the data generating process focuses on an increasingly small part of the unit interval over time, the width guarantees on our discretization will be insufficient to determine the distribution. Indeed, explicit constructions demonstrating the lack of sequential uniform convergence of linear threshold functions increasingly focus in this manner [Block et al., 2022].

Conversely, if $\forall t : \overline{\mathrm{CDF}}_t(v)$ was Lipschitz continuous in $v$, then our increasingly granular discretization would eventually overwhelm any fixed Lipschitz constant and guarantee uniform convergence. Theorem 3.3 expresses this intuition, but using the concept of smoothness rather than Lipschitz, as smoothness will allow us to generalize further [Rakhlin et al., 2011, Haghtalab et al., 2020, 2022b,a, Block et al., 2022].

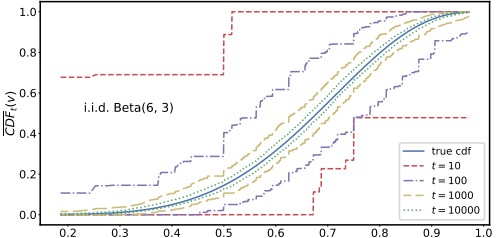
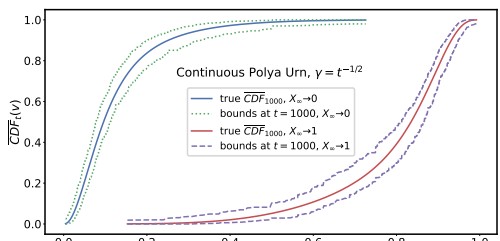

Figure 2: CDF bounds approaching the true CDF when sampling i.i.d. from a Beta(6,3) distribution. Note these bounds are simultaneously valid for all times and values.

Figure 3: Nonstationary Pólya simulation for two seeds approaching different average conditional CDFs. Bounds successfully track the true CDFs in both cases. See Section 4.2.

**Definition 3.2.** A distribution $D$ is $\xi$-smooth wrt reference measure $M$ if $D \ll M$ and $\operatorname{ess\,sup}_M \left( dD/dM \right) \leqslant \xi^{-1}$.

When the reference measure is the uniform distribution on the unit interval, $\xi$-smoothness implies an $\xi^{-1}$-Lipschitz CDF. However, when the reference measure has its own curvature, or charges points, the concepts diverge. When reading Theorem 3.3, note $\xi \leqslant 1$ (since the reference measure is a probability distribution) and as $\xi \to 0$ the smoothness constraint is increasingly relaxed. Thus Theorem 3.3 states "for less smooth distributions, convergence is slowed."

**Theorem 3.3.** *Let $U_t(v)$ and $L_t(v)$ be the upper and lower bounds returned by Algorithm 1 and Algorithm 2 respectively, when evaluated with $\epsilon(d) = 2^d$ and the confidence sequences $\Lambda_t$ and $\Xi_t$ of Equation* (15)*. If $\forall t : \mathbb{P}_t$ is $\xi_t$-smooth wrt the uniform distribution on the unit interval then*

$$\forall t, \forall v : U_t(v) - L_t(v) \leqslant$$

$$\sqrt{\frac{V_t}{t}} + \tilde{O}\left( \sqrt{\frac{V_t}{t} \log\left( \xi_t^{-2}\alpha^{-1}t^{3/2} \right)} \right), \tag{6}$$

*where $q_t \doteq \overline{\mathrm{CDF}}_t(v)$; $V_t \doteq 1/t + (q_t - 1/2)/\log(q_t/1-q_t)$; and $\tilde{O}()$ elides polylog $V_t$ factors.*

*Proof.* See Appendix C. $\qquad\square$

Theorem 3.3 matches our empirical results in two important aspects: (i) logarithmic dependence upon smoothness (e.g., Figure 4); (ii) tighter intervals for more extreme quantiles (e.g., Figure 2). Note the choice $\epsilon(d) = 2^d$ ensures the loop in Algorithm 1 terminates after at most $\log_2(\Delta)$ iterations, where $\Delta$ is the minimum difference between two distinct realized values.

**Worked Example**    To build intuition, in Appendix B.1 we explicitly calculate Algorithm 1 for a synthetic data set.

## 3.3   Extensions

**Arbitrary Support**    In Appendix D.1 we describe a variant of Algorithm 1 which uses a countable dense subset of the entire real line. It enjoys a similar guarantee to Theorem 3.3, but with an additional width which is logarithmic in the probe value $v$:  $\tilde{O}\left( \sqrt{\frac{V_t}{t} \log\left( \left(2 + \xi_t|v|t^{-1/2}\right)^2 \xi_t^{-2}\alpha^{-1}t^{3/2} \right)} \right)$. Note in this case $\xi_t$ is defined relative to (unnormalized) Lebesgue measure and can therefore exceed 1.

**Discrete Jumps**    If $\mathbb{P}_t$ is smooth wrt a reference measure which charges a countably infinite number of known discrete points, we can explicitly union bound over these additional points proportional to their density in the reference measure. In this case we preserve the above value-uniform guarantees. See Appendix D.2 for more details.

For distributions which charge unknown discrete points, we note the proof of Theorem 3.3 only exploits smoothness local to $v$. Therefore if the set of discrete points is nowhere dense, we eventually recover the guarantee of Equation (6) after a "burn-in" time $t$ which is logarithmic in the minimum distance from $v$ to a charged discrete point.

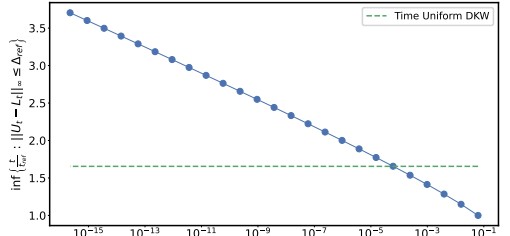
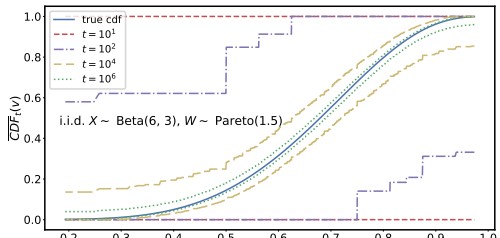

Figure 4: As smoothness $\epsilon$ decreases, we require more time to reach the same maximum confidence width. For low smoothness, DKW dominates our method. The logarithmic dependence matches our theory. See Section 4.1.

Figure 5: CDF bounds $v$ approaching the true counterfactual CDF when sampling i.i.d. from a Beta(6,3) with infinite-variance importance weights, using DDRM for the oracle confidence sequence.

### 3.4 Importance-Weighted Variant

An important use case is estimating a distribution based upon observations produced from another distribution with a known shift, e.g., arising in transfer learning [Pan and Yang, 2010] or off-policy evaluation [Waudby-Smith et al., 2022]. In this case the observations are tuples $(W_t, X_t)$, where the importance weight $W_t$ is a Radon-Nikodym derivative, implying $\forall t : \mathbb{E}_t[W_t] = 1$ and a.s. $W_t \geqslant 0$; and the goal is to estimate $\overline{\mathrm{CDF}}_t(v) = t^{-1} \sum_{s \leqslant t} \mathbb{E}_{s-1}[W_s 1_{X_s \leqslant v}]$. The basic approach in Algorithm 1 and Algorithm 2 is still applicable in this setting, but different $\Lambda_t$ and $\Xi_t$ are required. In Appendix E we present details on two possible choices for $\Lambda_t$ and $\Xi_t$: the first is based upon the empirical Bernstein construction of Howard et al. [2021], and the second based upon the DDRM construction of Mineiro [2022]. Both constructions leverage the $L^*$ Adagrad bound of Orabona [2019] to enable lazy evaluation. The empirical Bernstein version is amenable to analysis and computationally lightweight, but requires finite importance weight variance to converge (the variance bound need not be known, as the construction adapts to the unknown variance). The DDRM version requires more computation but produces tighter intervals. See Section 4.1 for a comparison.

Inspired by the empirical Bernstein variant, the following analog of Theorem 3.3 holds. Note $\mathbb{P}_t$ is the target (importance-weighted) distribution, not the observation (non-importance-weighted) distribution.

**Theorem 3.4.** *Let $U_t(v)$ and $L_t(v)$ be the upper and lower bounds returned by Algorithm 1 and Algorithm 2 respectively with $\epsilon(d) = 2^d$ and the confidence sequences $\Lambda_t$ and $\Xi_t$ of Equation* (18). *If $\forall t : \mathbb{P}_t$ is $\xi_t$-smooth wrt the uniform distribution on the unit interval then*

$$\forall t, \forall v : U_t(v) - L_t(v) \leqslant$$
$$B_t + \sqrt{\frac{(\tau + V_t)/t}{t}}$$
$$+ \tilde{O}\left(\sqrt{\frac{(\tau + V_t)/t}{t} \log\left(\xi_t^{-2} \alpha^{-1}\right)}\right) \tag{7}$$
$$+ \tilde{O}(t^{-1} \log\left(\xi_t^{-2} \alpha^{-1}\right)),$$

*where* $q_t \doteq \overline{\mathrm{CDF}}_t(v)$, $K(q_t) \doteq (q_t - 1/2)/\log(q_t/1-q_t)$; $V_t = O\left(K(q_t) \sum_{s \leqslant t} W_s^2\right)$, $B_t \doteq t^{-1} \sum_{s \leqslant t} (W_s - 1)$, *and* $\tilde{O}()$ *elides polylog* $V_t$ *factors.*

*Proof.* See Appendix E.2. ☐

Theorem 3.4 exhibits the following key properties: (i) logarithmic dependence upon smoothness; (ii) tighter intervals for extreme quantiles and importance weights with smaller quadratic variation; (iii) no explicit dependence upon importance weight range; (iv) asymptotic zero width for importance weights with sub-linear quadratic variation.

**Additional Remarks** First, the importance-weighted average CDF is a well-defined mathematical quantity, but the interpretation as a counterfactual distribution of outcomes given different actions in the controlled experimentation setting involves subtleties: we refer the interested reader to Waudby-Smith et al. [2022] for a complete discussion. Second, the need for nonstationarity techniques for

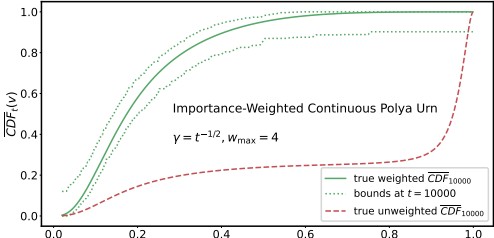
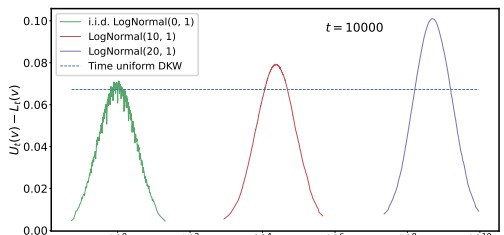

Figure 6: A nonstationary, importance-weighted simulation in which the factual distribution (red) diverges dramatically from the counterfactual distribution (green). The bound correctly covers the counterfactual CDF.

Figure 7: Demonstration of the variant described in Section 3.3 and Appendix D.1 for distributions with arbitrary support, based on i.i.d. sampling from a variety of lognormal distributions. Logarithmic range dependence is evident.

estimating the importance-weighted CDF is driven by the outcomes $(X_t)$ and not the importance-weights $(W_t)$. For example with off-policy contextual bandits, a changing historical policy does not induce nonstationarity, but a changing conditional reward distribution does.

## 4 Simulations

These simulations explore the empirical behaviour of Algorithm 1 and Algorithm 2 when instantiated with $\epsilon(d) = 2^d$ and curved boundary oracles $\Lambda$ and $\Xi$. To save space, precise details on the experiments as well additional figures are elided to Appendix F. Reference implementations which reproduce the figures are available at `https://github.com/microsoft/csrobust`.

### 4.1 The i.i.d. setting

These simulations exhibit our techniques on i.i.d. data. Although the i.i.d. setting does not fully exercise the technique, it is convenient for visualizing convergence to the unique true CDF. In this setting the DKW inequality applies, so to build intuition about our statistical efficiency, we compare our bounds with a naive time-uniform version of DKW resulting from a $(6/\pi^2 t^2)$ union bound over time.

**Beta distribution**   In this case the data is smooth wrt the uniform distribution on $[0, 1]$ so we can directly apply Algorithm 1 and Algorithm 2. Figure 2 shows the bounds converging to the true CDF as $t$ increases for an i.i.d. $\text{Beta}(6, 3)$ realization. Figure 8 compares the bound width to time-uniform DKW at $t = 10000$ for Beta distributions that are increasingly less smooth with respect to the uniform distribution. The DKW bound is identical for all, but our bound width increases as the smoothness decreases.

The additional figures in Appendix F clearly indicate tighter bounds at extreme quantiles, in correspondence with Theorem 3.3.

**Beyond the unit interval**   In Figure 7 (main text) and Appendix F.1 we present further simulations of i.i.d. lognormal and Gaussian random variables, ranging over $\mathbb{R}^+$ and $\mathbb{R}$ respectively, and using Algorithm 3. The logarithmic dependence of the bound width upon the probe value is evident.

**An Exhibition of Failure**   Figure 4 shows the (empirical) relative convergence when the data is simulated i.i.d. uniform over $[0, \epsilon]$ for decreasing $\epsilon$ (hence decreasing smoothness). The reference width is the maximum bound width obtained with Algorithm 1 and Algorithm 2 at $t_{\text{ref}} = 10000$ and $\epsilon = 1/16$, and shown is the multiplicative factor of time required for the maximum bound width to match the reference width as smoothness varies. The trend is consistent with arbitrarily poor convergence with arbitrarily small $\epsilon$. Because this is i.i.d. data, DKW applies and a uniform bound (independent of $\epsilon$) is available. Thus while our instance-dependent guarantees are valuable in practice, they can be dominated by stronger guarantees leveraging additional assumptions. On a positive note, a logarithmic dependence on smoothness is evident over many orders of magnitude, confirming the analysis of Theorem 3.3.

**Importance-Weighted**  In these simulations, in addition to being i.i.d., $X_t$ and $W_t$ are drawn independently of each other, so the importance weights merely increase the difficulty of ultimately estimating the same quantity.

In the importance-weighted case, an additional aspect is whether the importance-weights have finite or infinite variance. Figures 5 and 13 demonstrate convergence in both conditions when using DDRM for pointwise bounds. Figures 14 and 15 show the results using empirical Bernstein pointwise bounds. In theory, with enough samples and infinite precision, the infinite variance Pareto simulation would eventually cause the empirical Bernstein variant to reset to trivial bounds, but in practice this is not observed. Instead, DDRM is consistently tighter but also consistently more expensive to compute, as exemplified in Table 2. Thus either choice is potentially preferable.

Table 2: Comparison of DDRM and Empirical Bernstein on i.i.d. $X_t \sim \text{Beta}(6,3)$, for different $W_t$. Width denotes the maximum bound width $\sup_v U_t(v) - L_t(v)$. Time is for computing the bound at 1000 equally spaced points.

| $W_t$ | WHAT | WIDTH | TIME (SEC) |
|---|---|---|---|
| Exp(1) | DDRM | 0.09 | 24.8 |
| | EMP. BERN | 0.10 | 1.0 |
| Pareto($3/2$) | DDRM | 0.052 | 59.4 |
| | EMP. BERN | 0.125 | 2.4 |

### 4.2 Nonstationary

**Continuous Polya Urn**  In this case

$$X_t \sim \text{Beta}\left(2 + \gamma_t \sum_{s<t} 1_{X_s > 1/2}, 2 + \gamma_t \sum_{s<t} 1_{X_s \leqslant 1/2}\right),$$

i.e., $X_t$ is Beta distributed with parameters becoming more extreme over time: each realization will increasingly concentrate either towards 0 or 1. Suppose $\gamma_t = t^q$. In the most extreme case that $\left(t = \sum_{s \leqslant t} 1_{X_s > 1/2}\right)$, the conditional distribution at time $t$ is $\text{Beta}(x; 2 + t\gamma_t, 2) = O(t^{1+q})$, hence $d\mathbb{P}_t/dU = O(t^{1+q})$, which is smooth enough for our bounds to converge. Figure 3 shows the bounds covering the true CDF for two realizations with different limits. Figure 12 shows (for one realization) the maximum bound width, scaled by $\sqrt{t/\log(t)}$ to remove the primary trend, as a function of $t$ for different $\gamma_t$ schedules.

**Importance-Weighted Continuous Polya Urn**  In this case $W_t$ is drawn i.i.d. either $W_t = 0$ or $W_t = w_{\max}$, such as might occur during off-policy evaluation with an epsilon-greedy logging policy. Given $W_t$, the distribution of $X_t$ is given by

$$X_t | W_t \sim \text{Beta}\left(2 + \gamma_t \sum_{s<t} 1_{X_s > 1/2} 1_{W_s = W_t}, \right.$$

$$\left. 2 + \gamma_t \sum_{s<t} 1_{X_s < 1/2} 1_{W_s = W_t}\right),$$

i.e., each importance weight runs an independent Continuous Polya Urn. Because of this, it is possible for the unweighted CDF to mostly concentrate at one limit (e.g., 1) but the weighted CDF to concentrate at another limit (e.g., 0). Figure 6 exhibits this phenomenon.

## 5   Related Work

Constructing nonasymptotic confidence bands for the cumulative distribution function of i.i.d. random variables is a classical problem of statistical inference dating back to Dvoretzky et al. [1956] and Massart [1990]. While these bounds are quantile-uniform, they are ultimately fixed-time bounds

(i.e. not time-uniform). In other words, given a sample of i.i.d. random variables $X_1, \ldots, X_n \sim F$, these fixed time bounds $[\dot{L}_n(x), \dot{U}_n(x)]_{x \in \mathbb{R}}$ satisfy a guarantee of the form:

$$\mathbb{P}(\forall x \in \mathbb{R}, \ \dot{L}_n(x) \leqslant F(x) \leqslant \dot{U}_n(x)) \geqslant 1 - \alpha, \tag{8}$$

for any desired error level $\alpha \in (0, 1)$. Howard and Ramdas [2022] developed confidence bands $[\bar{L}_t(x), \bar{U}_t(x)]_{x \in \mathbb{R}, t \in \mathbb{N}}$ that are both quantile- *and* time-uniform, meaning that they satisfy the stronger guarantee:

$$\mathbb{P}(\forall x \in \mathbb{R}, t \in \mathbb{N}, \ \bar{L}_t(x) \leqslant F(x) \leqslant \bar{U}_t(x)) \geqslant 1 - \alpha. \tag{9}$$

However, the bounds presented in Howard and Ramdas [2022] ultimately focused on the classical i.i.d. *on-policy* setup, meaning the CDF for which confidence bands are derived is the same CDF as those of the observations $(X_t)_{t=1}^{\infty}$. This is in contrast to off-policy evaluation problems such as in randomized controlled trials, adaptive A/B tests, or contextual bandits, where the goal is to estimate a distribution different from that which was collected (e.g. collecting data based on a Bernoulli experiment with the goal of estimating the counterfactual distribution under treatment or control). Chandak et al. [2021] and Huang et al. [2021] both introduced fixed-time (i.e. non-time-uniform) confidence bands for the off-policy CDF in contextual bandit problems, though their procedures are quite different, rely on different proof techniques, and have different properties from one another. Waudby-Smith et al. [2022, Section 4] later developed *time-uniform* confidence bands in the off-policy setting, using a technique akin to Howard and Ramdas [2022, Theorem 5] and has several desirable properties in comparison to Chandak et al. [2021] and Huang et al. [2021] as outlined in Waudby-Smith et al. [2022, Table 2].

Nevertheless, regardless of time-uniformity or on/off-policy estimation, all of the aforementioned prior works assume that the distribution to be estimated is *fixed and unchanging over time*. The present paper takes a significant departure from the existing literature by deriving confidence bands that allow the distribution to change over time in a data-dependent manner, all while remaining time-uniform and applicable to off-policy problems in contextual bandits. Moreover, we achieve this by way of a novel stitching technique which is closely related to those of Howard and Ramdas [2022] and Waudby-Smith et al. [2022].

## 6 Discussion

This work constructs bounds by tracking specific values, in contrast with i.i.d. techniques which track specific quantiles. The value-based approach is amenable to proving correctness qua Theorem 3.1, but has the disadvantage of sensitivity to monotonic transformations. We speculate it is possible to be covariant to a fixed (wrt time) but unknown monotonic transformation without violating known impossibility results. A technique with this property would have increased practical utility.

## Acknowledgments and Disclosure of Funding

The authors thank Ian Waudby-Smith for insightful discussion and review.

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

# A  Confidence Sequences for Fixed $v$

Since our algorithm operates via reduction to pointwise confidence sequences, we provide a brief self-contained review here. We refer the interested reader to Howard et al. [2021] for a more thorough treatment.

A confidence sequence for a random process $X_t$ is a time-indexed collection of confidence sets $\mathrm{CI}_t$ with a time-uniform coverage property $\mathbb{P}\left(\forall t \in \mathbb{N} : X_t \in \mathrm{CI}_t\right) \geqslant 1 - \alpha$. For real random variables, the concept of a lower confidence sequence can be defined via $\mathbb{P}\left(\forall t \in \mathbb{N} : X_t \geqslant L_t\right) \geqslant 1 - \alpha$, and analogously for upper confidence sequences; and a lower and upper confidence sequence can be combined to form a confidence sequence $\mathrm{CI}_t \doteq \{x | L_t \leqslant x \leqslant U_t\}$ with coverage $(1 - 2\alpha)$ via a union bound.

One method for constructing a lower confidence sequence for a real valued parameter $z$ is to exhibit a real-valued random process $E_t(z)$ which, when evaluated at the true value $z^*$ of the parameter of interest, is a non-negative supermartingale with initial value of 1, in which case Ville's inequality ensures $\mathbb{P}\left(\forall t \in \mathbb{N} : E_t(z^*) \leqslant \alpha^{-1}\right) \geqslant 1 - \alpha$. If the process $E_t(z)$ is monotonically increasing in $z$, then the supremum of the lower contour set $L_t \doteq \sup_z \left\{z | E_t(z) \leqslant \alpha^{-1}\right\}$ is suitable as a lower confidence sequence; an upper confidence sequence can be analogously defined.

We use the above strategy as follows. We bound these deviations using the following nonnegative martingale,

$$E_t(\lambda) \doteq \exp\left(\lambda S_t - \sum_{s \leqslant t} \log\left(h(\lambda, \theta_s)\right)\right), \tag{10}$$

where $\lambda \in \mathbb{R}$ is fixed and $h(\lambda, z) \doteq (1 - z)e^{-\lambda z} + z e^{\lambda(1-z)}$, the moment-generating function of a centered Bernoulli($z$) random variable. Equation (10) is a test martingale qua Shafer et al. [2011], i.e., it can be used to construct time-uniform bounds on $\hat{q}_t - q_t$ via Ville's inequality.

Next we lower bound Equation (10),

$$E_t(\lambda) \doteq \exp\left(\lambda S_t - \sum_{s \leqslant t} \log\left(h(\lambda, \theta_s)\right)\right), \tag{10}$$

and eliminate the explicit dependence upon $\theta_s$, by noting $\log h(\lambda, \cdot)$ is concave and therefore

$$E_t(\lambda) \geqslant \exp\left(\lambda t\left(q_t - \hat{q}_t\right) - t\,\log h\left(\lambda, q_t\right)\right), \tag{11}$$

because $\left(t f(q) = \max_{\theta | 1^\top \theta = tq} \sum_{s \leqslant t} f(\theta_s)\right)$ for any concave $f$. Equation (11) is monotonically increasing in $q_t$ and therefore defines a lower confidence sequence. For an upper confidence sequence we use $q_t = 1 - (1 - q_t)$ and a lower confidence sequence on $(1 - q_t)$.

Regarding the choice of $\lambda$, in practice many $\lambda$ are (implicitly) used via stitching (i.e., using different $\lambda$ in different time epochs and majorizing the resulting bound in closed form) or mixing (i.e., using a particular fixed mixture of Equation (11) via a discrete sum or continuous integral over $\lambda$); our choices will depend upon whether we are designing for tight asymptotic rates or low computational footprint. We provide specific details associated with each theorem or experiment.

Note Equation (11) is invariant to permutations of $X_{1:t}$ and hence the empirical CDF at time $t$ is a sufficient statistic for calculating Equation (11) at any $v$.

## A.1  Challenge with quantile space

In this section assume all CDFs are invertible for ease of exposition.

In the i.i.d. setting, Equation (10) can be evaluated at the (unknown) fixed $v(q)$ which corresponds to quantile $q$. Without knowledge of the values, one can assert the existence of such values for a countably infinite collection of quantiles and a careful union bound of Ville's inequality on a particular discretization can yield an LIL rate: this is the approach of Howard and Ramdas [2022]. A key advantage of this approach is covariance to monotonic transformations.

Beyond the i.i.d. setting, one might hope to analogously evaluate Equation (10) at an unknown fixed value $v_t(q)$ which for each $t$ corresponds to quantile $q$. Unfortunately, $v_t(q)$ is not just unknown,

but also unpredictable with respect to the initial filtration, and the derivation that Equation (10) is a martingale depends upon $v$ being predictable. In the case that $X_t$ is independent but not identically distributed, $v_t(q)$ is initially predictable and therefore this approach could work, but would only be valid under this assumption.

The above argument does not completely foreclose the possibility of a quantile space approach, but merely serves to explain why the authors pursued a value space approach in this work. We encourage the interested reader to innovate.

# B Unit Interval Bounds

## B.1 Worked Example

Our (synthetic) data set consists of five values, each of which has occurred 1000 times:

$$D = \{(1000, 0), (1000, {}^1\!/\!7), (1000, {}^2\!/\!7), (1000, {}^3\!/\!7), (1000, {}^6\!/\!7)\}.$$

We use resolution $\epsilon(d) = 2^d$ and coverage error $\alpha = {}^1\!/\!20$.

**Upper bound for $v = {}^4\!/\!7$** The upper bound algorithm starts with the trivial upper bound of 1. The first evaluated point $\rho_1 = 2^{-1}\lceil 2^1 v \rceil = 1$ again yields the trivial bound of 1. There are still empirical counts between the probe value ($v = {}^4\!/\!7$) and the bound value ($\rho_1 = 1$) so the algorithm continues. The second evaluated point $\rho_2 = 2^{-2}\lceil 2^2 v \rceil = {}^3\!/\!4$, for which there are 4000 empirical counts below $\rho_2$ out of 5000 total. The pointwise confidence sequence is evaluated with counts $(4000, 5000)$ and coverage error $\delta_2 = {}^\alpha\!/\!24$, resulting in improved bound $\approx 0.825$. Now, there are no empirical counts between the probe value ($v = {}^4\!/\!7$) and the bound value ($\rho_2 = {}^3\!/\!4$) so the algorithm terminates. To see all subsequent bounds are dominated, note that a tighter upper bound $\rho_{d>2}$ will result in the same empirical counts (4000 out of 5000) but a looser coverage error and hence worse bound.

**Upper bound for $v = {}^{13}\!/\!28$** The upper bound algorithm starts with the trivial upper bound of 1. The first evaluated point $\rho_1 = 2^{-1}\lceil 2^1 v \rceil = {}^1\!/\!2$, for which there are 4000 empirical counts below $\rho_2$ out of 5000 total. The pointwise confidence sequence is evaluated with counts $(4000, 5000)$ and coverage error $\delta_1 = {}^\alpha\!/\!4$, resulting in improved bound $\approx 0.822$. There are no empirical counts between the probe value ($v = {}^{13}\!/\!28$) and the bound value ($\rho_1 = {}^1\!/\!2$) so the algorithm terminates. Relative to the previous example, the bound is slightly tighter as the discretization worked better for this $v$.

**Upper bound for $v = {}^5\!/\!14$** The upper bound algorithm starts with the trivial upper bound of 1. The first evaluated point $\rho_1 = 2^{-1}\lceil 2^1 v \rceil = {}^1\!/\!2$, for which there are 4000 empirical counts below $\rho_2$ out of 5000 total. The pointwise confidence sequence is evaluated with counts $(4000, 5000)$ and coverage error $\delta_1 = {}^\alpha\!/\!4$, resulting in improved bound $\approx 0.822$. There are empirical counts between the probe value ($v = {}^5\!/\!14$) and the bound value ($\rho_1 = {}^1\!/\!2$) so the algorithm continues. The second evaluated point $\rho_2 = 2^{-2}\lceil 2^2 v \rceil = {}^1\!/\!2$, which has the same empirical counts but worse coverage error at this level, and hence does not improve the bound. There are empirical counts between the probe value ($v = {}^5\!/\!14$) and the bound value ($\rho_2 = {}^1\!/\!2$) so the algorithm continues. The third evaluated point $\rho_3 = 2^{-3}\lceil 2^3 v \rceil = {}^3\!/\!8$, for which there are 3000 empirical counts below $\rho_3$ out of 5000 total. The pointwise confidence sequence is evaluated with counts $(3000, 5000)$ and coverage error $\delta_3 = {}^\alpha\!/\!96$, resulting in improved bound $\approx 0.633$. There are no empirical counts between the probe value ($v = {}^5\!/\!14$) and the bound value ($\rho_3 = {}^3\!/\!8$) so the algorithm terminates.

## B.2 Lower Bound

Algorithm 2 is extremely similar to Algorithm 1: the differences are indicated in comments. Careful inspection reveals the output of Algorithm 1, $U_t(v)$, can be obtained from the output of Algorithm 2, $L_t(v)$, via $U_t(v) = 1 - L_t(1 - v)$; but only if the sufficient statistics are adjusted such that $\Xi_t(\rho_d; \delta, \Psi_t) = 1 - \Lambda_t(1 - \rho_d; \delta, \Psi'_t)$. The reference implementation uses this strategy.

## B.3 Proof of Theorem 3.1

We prove the results for the upper bound Algorithm 1; the argument for the lower bound Algorithm 2 is similar.

---

**Algorithm 2** Unit Interval Lower Bound. $\epsilon(d)$ is an increasing function specifying the resolution of discretization at level $d$. $\Lambda_t\left(\rho; \delta, d, \Psi_t\right)$ is a lower confidence sequence for fixed value $\rho$ with coverage at least $(1 - \delta)$.

---

**Input:** value $v$; confidence $\alpha$; sufficient statistic $\Psi_t$.   *// comments below indicate differences from upper bound*
*// $\Psi_t \doteq X_{1:t}$ or $\Psi_t \doteq (W_{1:t}, X_{1:t})$*
**Output:** $L_t(v)$ satisfying Equation (2).
**if** $v < 0$ **then return** $0$ **end if**                    *// check for underflow of range rather than overflow*
$l \leftarrow 0$                                    *// initialize with 0 instead of 1*
$v \leftarrow \min(1, v)$                              *// project onto $[0, 1]$ using $\min$ instead of $\max$*
**for** $d = 1$ **to** $\infty$ **do**
  $\rho_d \leftarrow \epsilon(d)^{-1}\lfloor \epsilon(d) v \rfloor$                    *// use floor instead of ceiling*
  $\delta_d \leftarrow \alpha / 2^d \epsilon(d)$
  $l \leftarrow \max\left(l, \Lambda_t\left(\rho_d; \delta, \Psi_t\right)\right)$                *// use lower bound instead of upper bound*
  **if** $0 = \sum_{s \leqslant t} 1_{X_s \in [\rho_d, v)}$ **then**
    **return** $l$
  **end if**
**end for**

---

The algorithm terminates when we find a $d$ such that $0 = \sum_{s \leqslant t} 1_{X_s \in (v, \rho_d]}$. Since $\epsilon(d) \uparrow \infty$ as $d \uparrow \infty$, we have $\rho_d = \epsilon(d)\lceil \epsilon(d)^{-1} v \rceil \downarrow v$, so that $\sum_{s \leqslant t} 1_{X_s \in (v, \rho_d]} \downarrow 0$. So the algorithm must terminate.

At level $d$, we have $\epsilon(d)$ confidence sequences. The $i^{\text{th}}$ confidence sequence at level $d$ satisfies

$$P(\exists t : \overline{\mathrm{CDF}}_t(i/\epsilon(d)) > \Xi_t(i/\epsilon(d); \delta_d, d, \Psi_t)) \leqslant \frac{\alpha}{2^d \epsilon(d)}. \tag{12}$$

Taking a union bound over all confidence sequences at all levels, we have

$$P\left(\exists d \in \mathbb{N}, i \in \{1, \ldots, d\}, t \in \mathbb{N} : \overline{\mathrm{CDF}}_t(i/\epsilon(d)) > \Xi_t(i/\epsilon(d); \delta, d, \Psi_t)\right) \leqslant \alpha. \tag{13}$$

Thus we are assured that, for any $v \in \mathbb{R}$,

$$P(\forall t, d : \overline{\mathrm{CDF}}_t(v) \leqslant \overline{\mathrm{CDF}}_t(\rho_d) \leqslant \Xi_t(\rho_d; \delta_d, d, \Psi_t)) \geqslant 1 - \alpha. \tag{14}$$

Algorithm 1 will return $\Xi_t(\rho_d; \delta_d, d, \Psi_t)$ for some $d$ unless all such values are larger than one, in which case it returns the trivial upper bound of one. This proves the upper-bound half of guarantee (2). A similar argument proves the lower-bound half, and union bound over the upper and lower bounds finishes the argument.

## C   Proof of Theorem 3.3

**Theorem 3.3.** *Let $U_t(v)$ and $L_t(v)$ be the upper and lower bounds returned by Algorithm 1 and Algorithm 2 respectively, when evaluated with $\epsilon(d) = 2^d$ and the confidence sequences $\Lambda_t$ and $\Xi_t$ of Equation (15). If $\forall t : \mathbb{P}_t$ is $\xi_t$-smooth wrt the uniform distribution on the unit interval then*

$$\forall t, \forall v : U_t(v) - L_t(v) \leqslant$$
$$\sqrt{\frac{V_t}{t}} + \tilde{O}\left(\sqrt{\frac{V_t}{t} \log\left(\xi_t^{-2} \alpha^{-1} t^{3/2}\right)}\right), \tag{6}$$

*where $q_t \doteq \overline{\mathrm{CDF}}_t(v)$; $V_t \doteq 1/t + (q_t - 1/2)/\log(q_t/1 - q_t)$; and $\tilde{O}()$ elides polylog $V_t$ factors.*

Note $v$ is fixed for the entire argument below, and $\xi_t$ denotes the unknown smoothness parameter at time $t$.

We will argue that the upper confidence radius $U_t(v) - t^{-1} \sum_{s \leqslant t} 1_{X_s \leqslant v}$ has the desired rate. An analogous argument applies to the lower confidence radius $t^{-1} \sum_{s \leqslant t} 1_{X_s \leqslant v} - L_t(v)$, and the confidence width $U_t(v) - L_t(v)$ is the sum of these two.

For the proof we introduce an integer parameter $\eta \geqslant 2$ which controls both the grid spacing ($\epsilon(d) = \eta^d$) and the allocation of error probabilities to levels ($\delta_d = \alpha/(\eta^d \epsilon(d))$). In the main paper we set $\eta = 2$.

At level $d$ we construct $\eta^d$ confidence sequences on an evenly-spaced grid of values $1/\eta^d, 2/\eta^d, \ldots, 1$. We divide total error probability $\alpha/\eta^d$ at level $d$ among these $\eta^d$ confidence sequences, so that each individual confidence sequence has error probability $\alpha/\eta^{2d}$.

For a fixed bet $\lambda$ and value $\rho$, $S_t$ defined in Section 3.2 is sub-Bernoulli qua Howard et al. [2021, Definition 1] and therefore sub-Gaussian with variance process $V_t \doteq tK(q_t)$, where $K(p) \doteq (2p-1)/2\log(p/1-p)$ is from Kearns and Saul [1998]; from Howard et al. [2021, Proposition 5] it follows that there exists an explicit mixture distribution over $\lambda$ such that

$$M(t; q_t, \tau) \doteq \sqrt{2\left(tK(q_t) + \tau\right)\log\left(\frac{\eta^{2d}}{2\alpha}\sqrt{\frac{tK(q_t) + \tau}{\tau}} + 1\right)} \tag{15}$$

is a (curved) uniform crossing boundary, i.e., satisfies

$$\frac{\alpha}{\eta^{2d}} \geqslant \mathbb{P}\left(\exists t \geqslant 1 : S_t \geqslant \frac{M(t; q_t, \tau)}{t}\right),$$

where $S_t \doteq \overline{\mathrm{CDF}}_t(\rho) - t^{-1}\sum_{s\leqslant t} 1_{X_s \leqslant \rho}$ is from Equation (10), and $\tau$ is a hyperparameter to be determined further below.

Because the values at level $d$ are $1/\eta^d$ apart, the worst-case discretization error in the estimated average CDF value is

$$\overline{\mathrm{CDF}}_t(\epsilon(d)\lceil\epsilon(d)^{-1}v\rceil) - \overline{\mathrm{CDF}}_t(v) \leqslant 1/(\xi_t\eta^d),$$

and the total worst-case confidence radius including discretization error is

$$r_d(t) = \frac{1}{\xi_t\eta^d} + \sqrt{\frac{2\left(K(q_t) + \tau/t\right)}{t}\log\left(\frac{\eta^{2d}}{2\alpha}\sqrt{\frac{tK(q_t) + \tau}{\tau}} + 1\right)}.$$

Now evaluate at $d$ such that $\sqrt{\psi_t} < \xi_t\eta^d \leqslant \eta\sqrt{\psi_t}$ where $\psi_t \doteq t\left(K(q_t) + \tau/t\right)^{-1}$,

$$r_d(t) \leqslant \sqrt{\frac{K(q_t) + \tau/t}{t}} + \sqrt{\frac{2\left(K(q_t) + \tau/t\right)}{t}\log\left(\frac{\xi_t^{-2}\eta^2}{2\alpha}\left(\frac{t}{K(q_t) + \tau/t}\right)\sqrt{\frac{tK(q_t) + \tau}{\tau}} + 1\right)}.$$

The final result is not very sensitive to the choice of $\tau$, and we use $\tau = 1$ in practice.

# D   Extensions

## D.1   Arbitrary Support

Algorithm 3 is a variation on Algorithm 1 which does not assume a bounded range, and instead uses a countably discrete dense subset of the entire real line. Using the same argument of Theorem 3.3 with the modified probability from the modified union bound, we have

$$
\begin{aligned}
&|k_d| - 1 < \eta^{-d}|v| \leqslant |k_d|,\\
&\xi_t/\sqrt{\psi_t} > \eta^{-d} \geqslant \eta^{-1}\xi_t/\sqrt{\psi_t}\\
\implies\ &1 + |k_d| < 2 + \xi_t|v|/\sqrt{\psi_t}\\
\implies\ &r_d(t) \leqslant \tilde{O}\left(\sqrt{\frac{V_t}{t}\log\left(\left(2 + \xi_t|v|t^{-1/2}\right)^2 \xi_t^{-2}\alpha^{-1}t^{3/2}\right)}\right),
\end{aligned}
$$

demonstrating a logarithmic penalty in the probe value $v$ (e.g., Figure 7).

**Algorithm 3** Entire Real Line Upper Bound. $\epsilon(d)$ is an increasing function specifying the resolution of discretization at level $d$. $\Xi_t\left(\rho;\delta,d,\Psi_t\right)$ is an upper confidence sequence for fixed value $\rho$ with coverage at least $(1-\delta)$.

---

**Input:** value $v$; confidence $\alpha$; sufficient statistic $\Psi_t$.
// **e.g.** $\Psi_t \doteq X_{1:t}$ or $\Psi_t \doteq (W_{1:t}, X_{1:t})$
**Output:** $U_t(v)$ satisfying Equation (2).
$u \leftarrow 1$
**for** $d = 1$ **to** $\infty$ **do**
    $k_d \leftarrow \lceil \epsilon(d)^{-1}v \rceil$                                        // **Sub-optimal: see text for details**
    $\rho_d \leftarrow \epsilon(d)k_d$
    $\delta_d \leftarrow \left(\alpha/2^d\right)\left(3/(\pi^2-3)(1+|k_d|)^2\right)$          // **Union bound over** $d \in \mathbb{N}$ **and** $k_d \in \mathbb{Z}$
    $u \leftarrow \min\left(u, \Xi_t\left(\rho_d; \delta_d, d, \Psi_t\right)\right)$
    **if** $0 = \sum_{s \leqslant t} 1_{X_s \in (v, \rho_d]}$ **then**
        **return** $u$
    **end if**
**end for**

---

**Sub-optimality of** $k_d$   The choice of $k_d$ in Algorithm 3 is amenable to analysis, but unlike in Algorithm 1, it is not optimal. In Algorithm 1 the probability is allocated uniformly at each depth, and therefore the closest grid point provides the tightest estimate. However in Algorithm 3, the probability budget decreases with $|k_d|$ and because $k_d$ can be negative, it is possible that a different $k_d$ can produce a tighter upper bound. Since every $k_d$ is covered by the union bound, in principle we could optimize over all $k_d$ but it is unclear how to do this efficiently. In our implementation we do not search over all $k_d$, but we do adjust $k_d$ to be closest to the origin with the same empirical counts.

### D.2 Discrete Jumps

**Known Countably Infinite**   Suppose $D$ is smooth wrt a reference measure $M$, where $M$ is of the form

$$M = \breve{M} + \sum_{i \in I} \zeta_i 1_{v_i},$$

with $I$ a countable index set, $1 \geqslant \sum_{i \in I} \zeta_i$ and $\breve{M}$ a sub-probability measure normalizing to $(1 - \sum_{i \in I} \zeta_i)$. Then we can allocate $(1 - \sum_{i \in I} \zeta_i)$ of our overall coverage probability to bounding $\breve{M}$ using Algorithm 1 and Algorithm 2. For the remaining $\{v_i\}_{i \in I}$ we can run explicit pointwise bounds each with coverage probability fraction $\zeta_i$.

Computationally, early termination of the infinite search over the discrete bounds is possible. Suppose (wlog) $I$ indexes $\zeta$ in non-increasing order, i.e., $i \leqslant j \implies \zeta_i \leqslant \zeta_j$: then as soon as there are no remaining empirical counts between the desired value $v$ and the most recent discrete value $v_i$, the search over discrete bounds can terminate.

## E   Importance-Weighted Variant

### E.1   Modified Bounds

Algorithm 1 and Algorithm 2 are unmodified, with the caveat that the oracles $\Lambda_t$ and $\Xi_t$ must now operate on an importance-weighted realization $(W_{1:t}, X_{1:t})$, rather then directly on the realization $X_{1:t}$.

#### E.1.1   DDRM Variant

For simplicity we describe the lower bound $\Lambda_t$ only. The upper bound is derived analogously via the equality $Y_s = W_s - (W_s - Y_s)$ and a lower bound on $(W_s - Y_s)$: see Waudby-Smith et al. [2022, Remark 3] for more details.

This is the Heavy NSM from Mineiro [2022] combined with the $L^*$ bound of Orabona [2019, §4.2.3]. The Heavy NSM allow us to handle importance weights with unbounded variance, while the Adagrad $L^*$ bound facilitates lazy evaluation.

For fixed $v$, let $Y_t = W_t 1_{X_t \geqslant v}$ be a non-negative real-valued discrete-time random process, let $\hat{Y}_t \in [0, 1]$ be a predictable sequence, and let $\lambda \in [0, 1)$ be a fixed scalar bet. Then

$$E_t(\lambda) \doteq \exp\left(\lambda\left(\sum_{s \leqslant t} \hat{Y}_s - \mathbb{E}_{s-1}[Y_s]\right) + \sum_{s \leqslant t} \log\left(1 + \lambda\left(Y_s - \hat{Y}_s\right)\right)\right)$$

is a test supermartingale [Mineiro, 2022, §3]. Manipulating,

$$E_t(\lambda) = \exp\left(\lambda\left(\sum_{s \leqslant t} Y_s - \mathbb{E}_{s-1}[Y_s]\right) - \sum_{s \leqslant t} \underbrace{\left(\lambda\left(Y_s - \hat{Y}_s\right) - \log\left(1 + \lambda\left(Y_s - \hat{Y}_s\right)\right)\right)}_{\doteq h\left(\lambda\left(Y_s - \hat{Y}_s\right)\right)}\right)$$

$$= \exp\left(\lambda\left(\sum_{s \leqslant t} Y_s - \mathbb{E}_{s-1}[Y_s]\right) - \sum_{s \leqslant t} h\left(\lambda\left(Y_s - \hat{Y}_s\right)\right)\right)$$

$$\geqslant \exp\left(\lambda\left(\sum_{s \leqslant t} Y_s - \mathbb{E}_{s-1}[Y_s]\right) - \left(\sum_{s \leqslant t} h\left(\lambda\left(Y_s - \hat{Y}_t^*\right)\right)\right) - \text{Reg}(t)\right) \qquad (\dagger)$$

$$= \exp\left(\lambda\left(t\hat{Y}_t^* - \sum_{s \leqslant t} \mathbb{E}_{s-1}[Y_s]\right) + \sum_{s \leqslant t} \log\left(1 + \lambda\left(Y_s - \hat{Y}_t^*\right)\right) - \text{Reg}(t)\right),$$

where for $(\dagger)$ we use a no-regret learner on $h()$ with regret $\text{Reg}(t)$ to any constant prediction $\hat{Y}_t^* \in [0, 1]$. The function $h()$ is $M$-smooth with $M = \frac{\lambda^2}{(1-\lambda)^2}$ so we can get an $L^*$ bound [Orabona, 2019, §4.2.3] of

$$\text{Reg}(t) = 4\frac{\lambda^2}{(1-\lambda)^2} + 4\frac{\lambda}{1-\lambda}\sqrt{\sum_{s \leqslant t} h\left(\lambda\left(Y_s - \hat{Y}_t^*\right)\right)}$$

$$= 4\frac{\lambda^2}{(1-\lambda)^2} + 4\frac{\lambda}{1-\lambda}\sqrt{\left(-t\hat{Y}_t^* + \sum_{s \leqslant t} Y_s\right) - \sum_{s \leqslant t} \log\left(1 + \lambda\left(Y_s - \hat{Y}_t^*\right)\right)},$$

thus essentially our variance process is inflated by a square-root. In exchange we do not have to actually run the no-regret algorithm, which eases the computational burden. We can compete with any in-hindsight prediction: if we choose to compete with the clipped running mean $\overline{Y}_t$ then we end up with

$$E_t(\lambda) \geqslant \exp\left(\lambda\left(\min\left(t, \sum_{s \leqslant t} Y_s\right) - \mathbb{E}_{s-1}[Y_s]\right) + \sum_{s \leqslant t} \log\left(1 + \lambda\left(Y_s - \overline{Y}_t\right)\right) - \text{Reg}(t)\right),$$
(16)

which is implemented in the reference implementation as `LogApprox:getLowerBoundWithRegret(lam)`. The $\lambda$-s are mixed using DDRM from Mineiro [2022, Thm. 4], implemented via the `DDRM` class and the `getDDRMCSLowerBound` method in the reference implementation. `getDDRMCSLowerBound` provably correctly early terminates the infinite sum by leveraging

$$\sum_{s \leqslant t} \log\left(1 + \lambda\left(Y_s - \overline{Y}_t\right)\right) \leqslant \lambda\left(\sum_{s \leqslant t} Y_s - t\overline{Y}_t\right)$$

as seen in the termination criterion of the inner method `logwealth(mu)`.

To minimize computational overhead, we can lower bound $\log(a+b)$ for $b \geqslant 0$ using strong concavity qua Mineiro [2022, Thm. 3], resulting in the following geometrically spaced collection of sufficient statistics:

$$(1+k)^{n_l} = z_l \leqslant z < z_u = (1+k)z_l = (1+k)^{n_l+1},$$

along with distinct statistics for $z = 0$. $k$ is a hyperparameter controlling the granularity of the discretization (tighter lower bound vs. more space overhead): we use $k = 1/4$ exclusively in our

experiments. Note the coverage guarantee is preserved for any choice of $k$ since we are lower bounding the wealth.

Given these statistics, the wealth can be lower bounded given any bet $\lambda$ and any in-hindsight prediction $\hat{Y}_t^*$ via

$$f(z) \doteq \log \left( 1 + \lambda \left( z - \hat{Y}_t^* \right) \right),$$

$$f(z) \geqslant \alpha f(z_l) + (1 - \alpha) f(z_u) + \frac{1}{2} \alpha (1 - \alpha) m(z_l),$$

$$\alpha \doteq \frac{z_u - z}{z_u - z_l},$$

$$m(z_l) \doteq \left( \frac{k z_l \lambda}{k z_l \lambda + 1 - \lambda \hat{Y}_t^*} \right)^2 .$$

Thus when accumulating the statistics, for each $Y_s = W_s 1_{X_s \geqslant v}$, a value of $\alpha$ must be accumulated at key $f(z_l)$, a value of $(1 - \alpha)$ accumulated at key $f(z_u)$, and a value of $\alpha(1 - \alpha)$ accumulated at key $m(z_l)$. The `LogApprox::update` method from the reference implementation implements this.

Because these sufficient statistics are data linear, a further computational trick is to accumulate the sufficient statistics with equality only, i.e., for $Y_s = W_s 1_{X_s = v}$; and when the CDF curve is desired, combine these point statistics into cumulative statistics. In this manner only $O(1)$ incremental work is done per datapoint; while an additional $O(t \log(t))$ work is done to accumulate all the sufficient statistics only when the bounds need be computed. The method `StreamingDDRMECDF::Frozen::__init__` from the reference implementation contains this logic.

### E.1.2 Empirical Bernstein Variant

For simplicity we describe the lower bound $\Lambda_t$ only. The upper bound is derived analogously via the equality $Y_s = W_s - (W_s - Y_s)$ and a lower bound on $(W_s - Y_s)$: see Waudby-Smith et al. [2022, Remark 3] for more details.

This is the empirical Bernstein NSM from Howard et al. [2021] combined with the $L^*$ bound of Orabona [2019, §4.2.3]. Relative to DDRM it is faster to compute, has a more concise sufficient statistic, and is easier to analyze; but it is wider empirically, and theoretically requires finite importance weight variance to converge.

For fixed $v$, let $Y_t = W_t 1_{X_t \geqslant v}$ be a non-negative real-valued discrete-time random process, let $\hat{Y}_t \in [0, 1]$ be a predictable sequence, and let $\lambda \in [0, 1)$ be a fixed scalar bet. Then

$$E_t(\lambda) \doteq \exp \left( \lambda \left( \sum_{s \leqslant t} \hat{Y}_s - \mathbb{E}_{s-1}[Y_s] \right) + \sum_{s \leqslant t} \log \left( 1 + \lambda \left( Y_s - \hat{Y}_s \right) \right) \right)$$

is a test supermartingale [Mineiro, 2022, §3]. Manipulating,

$$E_t(\lambda) \doteq \exp \left( \lambda \left( \sum_{s \leqslant t} Y_s - \mathbb{E}_{s-1}[Y_s] \right) - \sum_{s \leqslant t} \underbrace{\left( \lambda \left( Y_s - \hat{Y}_s \right) - \log \left( 1 + \lambda \left( Y_s - \hat{Y}_s \right) \right) \right)}_{\doteq h\left( \lambda \left( Y_s - \hat{Y}_s \right) \right)} \right)$$

$$\geqslant \exp \left( \lambda \left( \sum_{s \leqslant t} Y_s - \mathbb{E}_{s-1}[Y_s] \right) - h(-\lambda) \sum_{s \leqslant t} \left( Y_s - \hat{Y}_s \right)^2 \right) \qquad \text{[Fan, Lemma 4.1]}$$

$$\geqslant \exp \left( \lambda \left( \sum_{s \leqslant t} Y_s - \mathbb{E}_{s-1}[Y_s] \right) - h(-\lambda) \left( \text{Reg}(t) + \sum_{s \leqslant t} (Y_s - Y_t^*)^2 \right) \right) \qquad (\dagger),$$

$$\doteq \exp \left( \lambda S_t - h(-\lambda) V_t \right),$$

where $S_t = \sum_{s \leqslant t} Y_s - \mathbb{E}_{s-1}[Y_s]$ and for $(\dagger)$ we use a no-regret learner on squared loss on feasible set $[0, 1]$ with regret $\text{Reg}(t)$ to any constant in-hindsight prediction $\hat{Y}_t^* \in [0, 1]$. Since $Y_s$ is unbounded

above, the loss is not Lipschitz and we can't get fast rates for squared loss, but we can run Adagrad and get an $L^*$ bound,

$$\text{Reg}(t) = 2\sqrt{2}\sqrt{\sum_{s \leqslant t} g_s^2}$$

$$= 4\sqrt{2}\sqrt{\sum_{s \leqslant t}(Y_s - \hat{Y}_s)^2}$$

$$\leqslant 4\sqrt{2}\sqrt{\text{Reg}(t) + \sum_{s \leqslant t}(Y_s - \hat{Y}_t^*)^2},$$

$$\implies \text{Reg}(t) \leqslant 16 + 4\sqrt{2}\sqrt{8 + \sum_{s \leqslant t}(Y_s - \hat{Y}_t^*)^2}.$$

Thus basically our variance process is inflated by an additive square root.

We will compete with $Y_t^* = \min\left(1, \frac{1}{t}\sum_s Y_s\right)$.

A key advantage of the empirical Bernstein over DDRM is the availability of both a conjugate (closed-form) mixture over $\lambda$ and a closed-form majorized stitched boundary. This yields both computational speedup and analytical tractability.

For a conjugate mixture, we use the truncated gamma prior from Waudby-Smith et al. [2022, Theorem 2] which yields mixture wealth

$$M_t^{\text{EB}} \doteq \left(\frac{\tau^\tau e^{-\tau}}{\Gamma(\tau) - \Gamma(\tau, \tau)}\right)\left(\frac{1}{\tau + V_t}\right){}_1F_1\left(1, V_t + \tau + 1, S_t + V_t + \tau\right), \quad (17)$$

where ${}_1F_1(\ldots)$ is Kummer's confluent hypergeometric function and $\Gamma(\cdot, \cdot)$ is the upper incomplete gamma function. For the hyperparameter, we use $\tau = 1$.

### E.2 Proof of Theorem 3.4

**Theorem 3.4.** *Let $U_t(v)$ and $L_t(v)$ be the upper and lower bounds returned by Algorithm 1 and Algorithm 2 respectively with $\epsilon(d) = 2^d$ and the confidence sequences $\Lambda_t$ and $\Xi_t$ of Equation* (18)*. If $\forall t : \mathbb{P}_t$ is $\xi_t$-smooth wrt the uniform distribution on the unit interval then*

$$\forall t, \forall v : U_t(v) - L_t(v) \leqslant$$
$$B_t + \sqrt{\frac{(\tau + V_t)/t}{t}}$$
$$+ \tilde{O}\left(\sqrt{\frac{(\tau + V_t)/t}{t}}\log\left(\xi_t^{-2}\alpha^{-1}\right)\right) \quad (7)$$
$$+ \tilde{O}(t^{-1}\log\left(\xi_t^{-2}\alpha^{-1}\right)),$$

*where $q_t \doteq \overline{\text{CDF}}_t(v)$, $K(q_t) \doteq {}^{(q_t - 1/2)}/_{\log(q_t/1 - q_t)}$; $V_t = O\left(K(q_t)\sum_{s \leqslant t} W_s^2\right)$, $B_t \doteq t^{-1}\sum_{s \leqslant t}(W_s - 1)$, and $\tilde{O}()$ elides polylog $V_t$ factors.*

Note $v$ is fixed for the entire argument below, and $\xi_t$ denotes the unknown smoothness parameter at time $t$.

We will argue that the upper confidence radius $U_t(v) - t^{-1}\sum_{s \leqslant t} W_s 1_{X_s \leqslant v}$ has the desired rate. An analogous argument applies to the lower confidence radius. One difference from the non-importance-weighted case is that, to be sub-exponential, the lower bound is constructed from an upper bound on $U_t'(v) = W_s(1 - 1_{X_s \leqslant v})$ via $L_t(v) - 1 - U_t'(v)$, which introduces an additional $B_t = t^{-1}\sum_{s \leqslant t}(W_s - 1)$ term to the width. (Note, because $\forall t : \mathbb{E}_t[W_t - 1] = 0$, this term will concentrate, but we will simply use the realized value here.)

For the proof we introduce an integer parameter $\eta \geqslant 2$ which controls both the grid spacing ($\epsilon(d) = \eta^d$) and the allocation of error probabilities to levels ($\delta_d = \alpha/(\eta^d\epsilon(d))$). In the main paper we set $\eta = 2$.

At level $d$ we construct $\eta^d$ confidence sequences on an evenly-spaced grid of values $1/\eta^d, 2/\eta^d, \ldots, 1$. We divide total error probability $\alpha/\eta^d$ at level $d$ among these $\eta^d$ confidence sequences, so that each individual confidence sequence has error probability $\alpha/\eta^{2d}$.

For a fixed bet $\lambda$ and value $\rho$, $S_t$ defined in Appendix E.1.2 is sub-exponential qua Howard et al. [2021, Definition 1] and therefore from Lemma E.1 there exists an explicit mixture distribution over $\lambda$ inducing (curved) boundary

$$\frac{\alpha}{\eta^{2d}} \geqslant \mathbb{P}\left(\exists t \geqslant 1 : \frac{S_t}{t} \geqslant \max\left(\frac{C(\tau)}{t}, u\left(V_t; \tau, \frac{\alpha}{\eta^{2d}}\right)\right)\right),$$

$$u\left(V_t; \tau, \frac{\alpha}{\eta^{2d}}\right) = \sqrt{2\left(\frac{(\tau + V_t)/t}{t}\right) \log\left(\sqrt{\frac{\tau + V_t}{2\pi}} e^{-\frac{1}{12(\tau + V_t)+1}} \left(\frac{1 + \eta^{2d}\alpha^{-1}}{C(\tau)}\right)\right)}$$

$$+ \frac{1}{t}\log\left(\sqrt{\frac{\tau + V_t}{2\pi}} e^{-\frac{1}{12(\tau + V_t)+1}} \left(\frac{1 + \eta^{2d}\alpha^{-1}}{C(\tau)}\right)\right), \tag{18}$$

where $S_t \doteq \overline{\text{CDF}}_t(\rho) - t^{-1}\sum_{s \leqslant t} W_s 1_{X_s \leqslant \rho}$, and $\tau$ is a hyperparameter to be determined further below.

Because the values at level $d$ are $1/\eta^d$ apart, the worst-case discretization error in the estimated average CDF value is

$$\overline{\text{CDF}}_t(\epsilon(d)\lceil\epsilon(d)^{-1}v\rceil) - \overline{\text{CDF}}_t(v) \leqslant 1/(\xi_t\eta^d),$$

and the total worst-case confidence radius including discretization error is

$$r_d(t) = \frac{1}{\xi_t\eta^d} + \max\left(\frac{C(\tau)}{t}, u\left(V_t; \tau, \frac{\alpha}{\eta^{2d}}\right)\right).$$

Now evaluate at $d$ such that $\sqrt{\psi_t} < \xi_t\eta^d \leqslant \eta\sqrt{\psi_t}$ where $\psi_t \doteq t\left((\tau + V_t)/t\right)^{-1}$,

$$r_d(t) \leqslant \frac{1}{\sqrt{\psi_t}} + \max\left(\frac{C(\tau)}{t}, u\left(V_t; \tau, \frac{\alpha}{\eta^2\xi_t^{-2}\psi_t}\right)\right)$$

$$= \sqrt{\frac{(\tau + V_t)/t}{t}} + \tilde{O}\left(\sqrt{\frac{(\tau + V_t)/t}{t}\log\left(\xi_t^{-2}\alpha^{-1}\right)}\right) + \tilde{O}(t^{-1}\log\left(\xi_t^{-2}\alpha^{-1}\right)),$$

where $\tilde{O}()$ elides polylog $V_t$ factors. The final result is not very sensitive to the choice of $\tau$, and we use $\tau = 1$ in practice.

**Lemma E.1.** *Suppose*

$$\exp\left(\lambda S_t - \psi_e(\lambda)V_t\right),$$
$$\psi_e(\lambda) \doteq -\lambda - \log(1 - \lambda),$$

*is sub-$\psi_e$ qua Howard et al. [2021, Definition 1]; then there exists an explicit mixture distribution over $\lambda$ with hyperparameter $\tau > 0$ such that*

$$\alpha \geqslant \mathbb{P}\left(\exists t \geqslant 1 : \frac{S_t}{t} \geqslant \max\left(\frac{C(\tau)}{t}, u\left(V_t; \tau, \alpha\right)\right)\right),$$

$$u\left(V_t; \tau, \alpha\right) = \sqrt{2\left(\frac{(\tau + V_t)/t}{t}\right)\log\left(\sqrt{\frac{\tau + V_t}{2\pi}}e^{-\frac{1}{12(\tau + V_t)+1}}\left(\frac{1 + \alpha^{-1}}{C(\tau)}\right)\right)}$$

$$+ \frac{1}{t}\log\left(\sqrt{\frac{\tau + V_t}{2\pi}}e^{-\frac{1}{12(\tau + V_t)+1}}\left(\frac{1 + \alpha^{-1}}{C(\tau)}\right)\right),$$

$$C(\tau) \doteq \frac{\tau^\tau e^{-\tau}}{\Gamma(\tau) - \Gamma(\tau, \tau)},$$

*is a (curved) uniform crossing boundary.*

*Proof.* We can form the conjugate mixture using a truncated gamma prior from Howard et al. [2021, Proposition 9], in the form from Waudby-Smith et al. [2022, Theorem 2], which is our Equation (17).

$$M_t^{\text{EB}} \doteq \left( \frac{\tau^\tau e^{-\tau}}{\Gamma(\tau) - \Gamma(\tau, \tau)} \right) \left( \frac{1}{\tau + V_t} \right) {}_1F_1 \left( 1, V_t + \tau + 1, S_t + V_t + \tau \right),$$

where ${}_1F_1(\dots)$ is Kummer's confluent hypergeometric function. Using Olver et al. [2010, identity 13.6.5],

$$ {}_1F_1(1, a + 1, x) = e^x a x^{-a} \left( \Gamma(a) - \Gamma(a, x) \right) $$

where $\Gamma(a, x)$ is the (unregularized) upper incomplete gamma function. From Pinelis [2020, Theorem 1.2] we have

$$\Gamma(a, x) < \frac{x^a e^{-x}}{x - a}$$

$$\implies {}_1F_1(1, a + 1, x) \geqslant e^x a x^{-a} \Gamma(a) - \frac{a}{x - a}.$$

Applying this to the mixture yields

$$M_t^{\text{EB}} \geqslant \frac{C(\tau) e^{\tau + V_t + S_t}}{(\tau + V_t + S_t)^{\tau + V_t}} \Gamma(\tau + V_t) - \frac{C(\tau)}{S_t}$$

$$\geqslant \frac{C(\tau) e^{\tau + V_t + S_t}}{(\tau + V_t + S_t)^{\tau + V_t}} \Gamma(\tau + V_t) - 1, \tag{$\dagger$}$$

where ($\dagger$) follows from the self-imposed constraint $S_t \geqslant C(\tau)$. This yields crossing boundary

$$\alpha^{-1} = \frac{C(\tau) e^{\tau + V_t + S_t}}{(\tau + V_t + S_t)^{\tau + V_t}} \Gamma(\tau + V_t) - 1,$$

$$\frac{e^{\tau + V_t + S_t}}{\left( 1 + \frac{S_t}{\tau + V_t} \right)^{\tau + V_t}} = \left( \frac{(\tau + V_t)^{\tau + V_t}}{\Gamma(\tau + V_t)} \right) \left( \frac{1 + \alpha^{-1}}{C(\tau)} \right) \doteq \left( \frac{(\tau + V_t)^{\tau + V_t}}{\Gamma(\tau + V_t)} \right) \phi_t(\tau, \alpha),$$

$$\frac{e^{1 + \frac{S_t}{\tau + V_t}}}{\left( 1 + \frac{S_t}{\tau + V_t} \right)} = \left( \frac{(\tau + V_t)^{\tau + V_t}}{\Gamma(\tau + V_t)} \right)^{\frac{1}{\tau + V_t}} \phi_t(\tau, \alpha)^{\frac{1}{\tau + V_t}} \doteq z_t,$$

$$S_t = (\tau + V_t) \left( -1 - W_{-1} \left( -z_t^{-1} \right) \right).$$

Chatzigeorgiou [2013, Theorem 1] states

$$W_{-1}(-e^{-u-1}) \in -1 - \sqrt{2u} + \left[ -u, -\frac{2}{3} u \right]$$

$$\implies -1 - W_{-1}(-e^{-u-1}) \in \sqrt{2u} + \left[ \frac{2}{3} u, u \right].$$

Substituting yields

$$(\tau + V_t) \left( -1 - W_{-1} \left( -z_t^{-1} \right) \right) \leqslant (\tau + V_t) \left( \sqrt{2 \log \left( \frac{z_t}{e^1} \right)} + \log \left( \frac{z_t}{e^1} \right) \right). \tag{19}$$

From Feller [1958, Equation (9.8)] we have

$$\Gamma(1 + n) \in \sqrt{2\pi n} \left( \frac{n}{e^1} \right)^n \left[ e^{\frac{1}{12n+1}}, e^{\frac{1}{12n}} \right]$$

$$\implies \left( \frac{(\tau + V_t)^{\tau + V_t}}{\Gamma(\tau + V_t)} \right)^{\frac{1}{\tau + V_t}} \in \left( \frac{\tau + V_t}{2\pi} \right)^{\frac{1}{2(\tau + V_t)}} e^1 \left[ e^{-\frac{1}{12(\tau + V_t)^2}}, e^{-\frac{1}{12(\tau + V_t)^2 + (\tau + V_t)}} \right].$$

Therefore

$$(\tau + V_t) \sqrt{2 \log \left( \frac{z_t}{e^1} \right)} \leqslant (\tau + V_t) \sqrt{2 \log \left( \left( \frac{\tau + V_t}{2\pi} \right)^{\frac{1}{2(\tau + V_t)}} e^{-\frac{1}{12(\tau + V_t)^2 + (\tau + V_t)}} \phi_t(\tau, \alpha)^{\frac{1}{\tau + V_t}} \right)}$$

$$= \sqrt{2 (\tau + V_t) \log \left( \sqrt{\frac{\tau + V_t}{2\pi}} e^{-\frac{1}{12(\tau + V_t)+1}} \phi_t(\tau, \alpha) \right)}, \tag{20}$$

and

$$(\tau + V_t)\log\left(\frac{z_t}{e^1}\right) \leqslant (\tau + V_t)\log\left(\left(\frac{\tau + V_t}{2\pi}\right)^{\frac{1}{2(\tau+V_t)}} e^{-\frac{1}{12(\tau+V_t)^2+(\tau+V_t)}}\phi_t(\tau,\alpha)^{\frac{1}{\tau+V_t}}\right)$$

$$= \log\left(\sqrt{\frac{\tau + V_t}{2\pi}}e^{-\frac{1}{12(\tau+V_t)+1}}\phi_t(\tau,\alpha)\right). \tag{21}$$

Combining Equations (19) to (21) yields the crossing boundary

$$\frac{S_t}{t} = \sqrt{2\left(\frac{(\tau+V_t)/t}{t}\right)\log\left(\sqrt{\frac{\tau+V_t}{2\pi}}e^{-\frac{1}{12(\tau+V_t)+1}}\left(\frac{1+\alpha^{-1}}{C(\tau)}\right)\right)}$$
$$+ \frac{1}{t}\log\left(\sqrt{\frac{\tau+V_t}{2\pi}}e^{-\frac{1}{12(\tau+V_t)+1}}\left(\frac{1+\alpha^{-1}}{C(\tau)}\right)\right).$$

$\square$

# F   Simulations

## F.1   i.i.d. setting

For non-importance-weighted simulations, we use the Beta-Binomial boundary of Howard et al. [2021] for $\Lambda_t$ and $\Xi_t$. The curved boundary is induced by the test NSM

$$W_t(b;\hat{q}_t, q_t) = \frac{\int_{q_t}^1 d\text{Beta}\left(p; bq_t, b(1-q_t)\right)\left(\frac{p}{q_t}\right)^{t\hat{q}_t}\left(\frac{1-p}{1-q_t}\right)^{t(1-\hat{q}_t)}}{\int_{q_t}^1 d\text{Beta}\left(p; bq_t, b(1-q_t)\right)}$$

$$= \frac{1}{(1-q_t)^{t(1-\hat{q}_t)}q_t^{t\hat{q}_t}}\left(\frac{\text{Beta}(q_t, 1, bq_t + t\hat{q}_t, b(1-q_t) + t(1-\hat{q}_t))}{\text{Beta}(q_t, 1, bq_t, b(1-q_t))}\right)$$

with prior parameter $b = 1$. Further documentation and details are in the reference implementation `csnsquantile.ipynb`.

The importance-weighted simulations use the constructions from Appendix E: the reference implementation is in `csnsopquantile.ipynb` for the DDRM variant and `csnsopquantile-ebern.ipynb` for the empirical Bernstein variant.

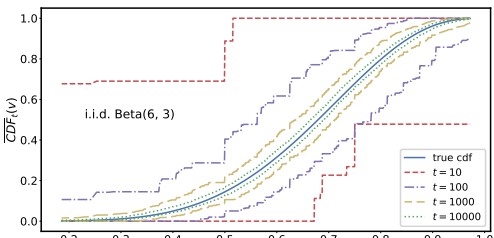

Figure 2: CDF bounds approaching the true CDF when sampling i.i.d. from a Beta(6,3) distribution. Note these bounds are simultaneously valid for all times and values.

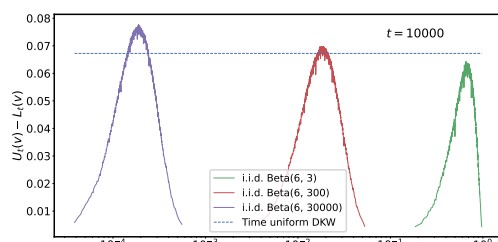

Figure 8: Comparison to naive time-uniform DKW (which is only valid in the i.i.d. setting) for Beta distributions of varying smoothness. Decreasing smoothness degrades our bound.

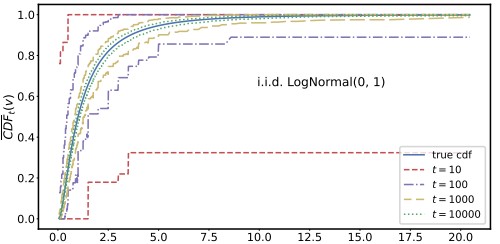

Figure 9: CDF bounds approaching the true CDF when sampling i.i.d. from a lognormal(0, 1) distribution. Recall these bounds are simultaneously valid for all times and values.

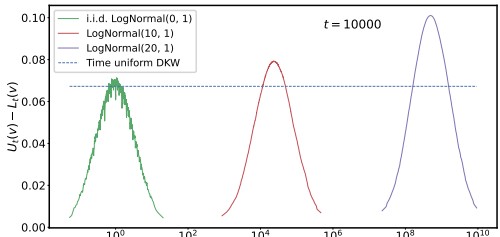

Figure 7: Demonstration of the variant described in Section 3.3 and Appendix D.1 for distributions with arbitrary support, based on i.i.d. sampling from a variety of lognormal distributions. Logarithmic range dependence is evident.

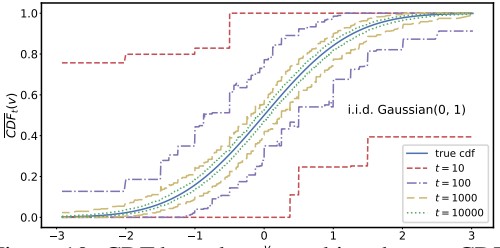

Figure 10: CDF bounds approaching the true CDF when sampling i.i.d. from a Gaussian(0, 1) distribution. Recall these bounds are simultaneously valid for all times and values.

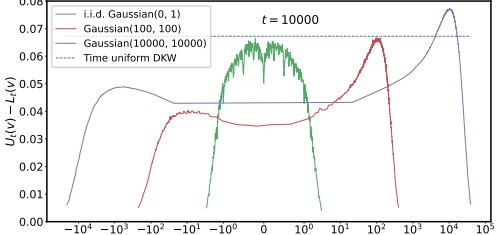

Figure 11: Demonstration of the variant described in Section 3.3 and Appendix D.1 for distributions with arbitrary support, based on i.i.d. sampling from a variety of Gaussian distributions. Logarithmic range dependence is evident.

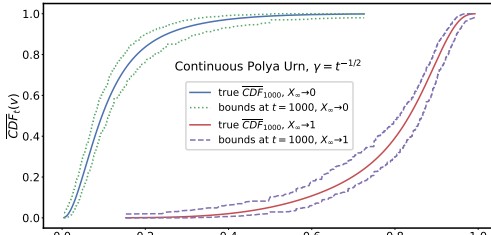

Figure 3: Nonstationary Polya simulation for two seeds approaching different average conditional CDFs. Bounds successfully track the true CDFs in both cases. See Section 4.2.

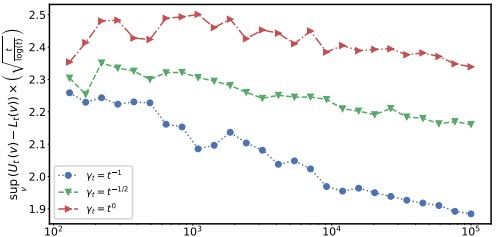

Figure 12: Maximum bound width, scaled by $\sqrt{t/\log(t)}$ to remove the primary trend, as a function of $t$, for nonstationary Polya simulations with different $\gamma_t$ schedules. See Section 4.2

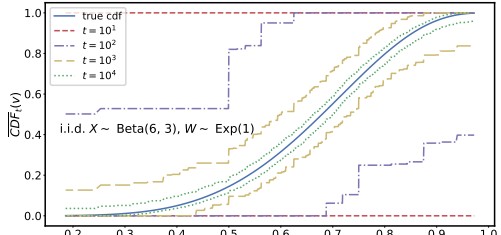

Figure 13: CDF bounds approaching the true counterfactual CDF when sampling i.i.d. from a Beta(6,3) with finite-variance importance weights, using DDRM for the oracle confidence sequence.

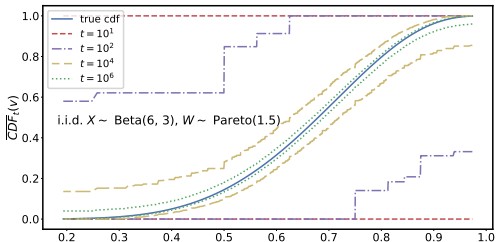

Figure 5: CDF bounds approaching the true counterfactual CDF when sampling i.i.d. from a Beta(6,3) with infinite-variance importance weights, using DDRM for the oracle confidence sequence.

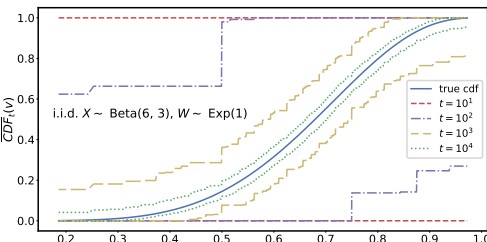

Figure 14: CDF bounds approaching the true counterfactual CDF when sampling i.i.d. from a Beta(6,3) with finite-variance importance weights, using Empirical Bernstein for the oracle confidence sequence.

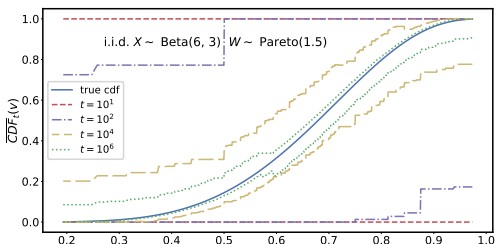

Figure 15: CDF bounds approaching the true counterfactual CDF when sampling i.i.d. from a Beta(6,3) with infinite-variance importance weights, using Empirical Bernstein for the oracle confidence sequence. Despite apparent convergence, eventually this simulation would reset the Empirical Bernstein oracle confidence sequence to trivial bounds.

