# OpenReview forum: "Time-uniform confidence bands for the CDF under nonstationarity"
_NeurIPS.cc/2023/Conference — NeurIPS 2023 poster_

### Official Review · Reviewer_gyN9 · 2023-06-27

**Soundness:** 2 fair
**Presentation:** 4 excellent
**Contribution:** 3 good
**Rating:** 6
**Confidence:** 3

**Summary:**

The authors propose a method for constructing time-uniform and value-uniform bounds on the CDF of the averaged historical distribution of a real valued random process. They propose a design principle AVAST (always valid and sometimes trivial), implying that their bounds always have coverage, but may converge to 0 slowly. The authors build upon the work of [Howard and Ramdas 2022], extending those results to non-stationary processes, and allowing for counterfactual estimation.

**Strengths:**

1. The authors provide a new algorithm for constructing time and value uniform bounds on the CDF of the averaged historical distribution of a real valued random process in the non-stationary case. The theoretical guarantees are cleanly provided, and the proofs appear to be correct.
2. The authors extend these results to importance-weighted random variables.
3. The authors simulate their algorithm on several synthetic examples, showcasing its adaptive performance.

**Weaknesses:**

The one point that is not clear to me is the instance-adaptivity of Theorem 3.3. Is this a constant bound $\xi_t$ constant for all $t$, i.e. $\xi_t \le \xi$ for all $t$? If not, how is it that the bound is time-uniform while having no dependence on previous $\xi_t$? It seems as though if $X_1,\dots,X_{100}$ were i.i.d. $U[0,\epsilon]$, and $X_{101}$ is independent $U[0,1]$, then the confidence interval width $U_{101}(\epsilon/2)-L_{101}(\epsilon/2)$ would depend critically on $\epsilon$. Simulations would also be helpful to clarify this point.

**Questions:**

See weaknesses, the instance-adaptivity of Theorem 3.3 is unclear. This work seems polished and well-written aside from this, but this result seems like a critical part of the paper. I am leaving a score of weak reject, but if this point is clarified I would be happy to see this paper accepted.

**Limitations:**

Line 79: leveraging *the* monotonicity
line 137: two commas after e.g.
Line 427: For a fixed "bet" $\lambda$. It is not explained why $\lambda$ should be treated as a bet.

---

> ### Author Rebuttal · Authors · 2023-08-07
>
> > The one point that is not clear to me is the instance-adaptivity of Theorem 3.3.
>
> Thanks for pointing this out, we'll add more commentary, since this is an important and desirable property of the technique which should be highlighted.
>
> The width guarantee is dependent upon the smoothness of the averaged historical distribution up to time t, which can vary with t (as the technique adapts to the [unknown] smoothness at each timestep).  In particular the distributions can become polynomial (in t) less smooth over time and we still enjoy a 1/sqrt(t) shrinking width up to log(t) factors.  It is only when distributions are exponentially less smooth with time that our width becomes constant, which (not coincidentally) is what the impossibility construction of Block et. al. does.
>
> > For a fixed "bet" $\lambda$. It is not explained why $\lambda$ should be treated as a bet.
>
> Thanks for pointing this out.  When we moved material from the main text to the appendix, we lost the definition of $S_t$ (as another reviewer pointed out).  Defining that will clarify.  Furthermore the word "bet" is a term of art from the confidence sequence literature which is not helpful here, so we'll switch to using the word "parameter".

---

> > ### Comment · Reviewer_gyN9 · 2023-08-10
> > **Response**
> >
> > Thank you for the clarification; I now realize that I had misread notation, and that $\mathbb{P}_t$ is the $\textit{averaged historical distribution}$, as opposed to the distribution of $X_t$ itself. My apologies. This seems to be somewhat confusing notationally, as the CDF is already defined in terms of the averaged historical distribution.
> >
> > In light of this, I am increasing my score to a 6; I think this paper should be accepted, but will not champion it. The contribution of the paper is clear, the problem it tackles is interesting, and the theoretical results it provides are useful, but the writing and clarity could definitely be improved (as shown by the questions of the other reviewers).

---

### Official Review · Reviewer_jBxN · 2023-07-06

**Soundness:** 3 good
**Presentation:** 2 fair
**Contribution:** 3 good
**Rating:** 5
**Confidence:** 2

**Summary:**

The paper considers the problem of estimating the CDF of a sequence of random variables obtained sequentially according to some distributions.
Many classical results are known for this problem when the random variable sequence is obtained in an i.i.d. manner following a certain distribution, along with its confidence bands.
On the other hand, some impossibility results are known for the case where the random variables are dependent on each other, and their confidence bands are not sufficiently known.
To address this issue, the paper constructs confidence bands for CDFs with running averaged conditional distributions that are uniform in time and value.
The extension to importance-weighted random variables is also discussed.

**Strengths:**

- The paper shows how to construct confidence bands in the setting where there is dependence in the sequence of random variables obtained, an important problem in practical applications. This result is further extended to the importance-weighted setting, which is frequently encountered in many online decision making problems.
- Through a number of experimental results, the paper shows that results that are consistent with the theoretical results are obtained.

**Weaknesses:**

- The contents of the Introduction seem to be insufficient, because it is not clear what kind of problem is being solved just by following line 12 to line 27, and then the explanation of contirbutions is given immediately after that.
Section 5 (Related Work) seems to be useful for understanding the problem the authors are trying to solve, so it would be desirable to include it earlier. Even after adding it, the explanation of background knowledge still seems insufficient, so it seems desirable to use the remaining pages to provide more explanation in the current Introduction.

- Although the comparison with existing results is clearly stated, the reviewer could not tell from the text what the major technical difficulties of the problems are. Many of the proofs and contributions are included in the appendix, and it would be desirable to have an explanation of what technical difficulties were solved to obtain the main results, even if they are only sketches.

Minor issues and typos:
- The horizontal line labels in Figures 1 to 7 are too small to see.

**Questions:**

- As noted in Weaknesses, there are few specific descriptions of technical difficulties, and reviewers expect authors to address them.

---

> ### Author Rebuttal · Authors · 2023-08-07
>
> > The contents of the Introduction seem to be insufficient
>
> Great point, the current introduction is abstract and a concrete example will be helpful.  We will discuss the scenario of continuous monitoring for software regression detection, which combines the desire to do inference beyond the mean (hence the entire CDF) with a desire to detect changes as rapidly as possible in a nonstationary environment.[[1]](https://arxiv.org/abs/2205.14762)
>
> > Many of the proofs and contributions are included in the appendix, and it would be desirable to have an explanation of what technical difficulties were solved to obtain the main results, even if they are only sketches.
>
> Thanks for pointing this out.  We'll provide short sketches of the proofs within the main text.  For example, for the first two theorems:
> * Theorem 3.1 follows from the guarantees of the confidence sequences at each fixed value, the monotonicity of the CDF, and a union bound.
> * Theorem 3.3 follows by noting the bias of the discretization is bounded by the smoothness, and then propagating that through a closed-form confidence boundary.
>
> > The horizontal line labels in Figures 1 to 7 are too small to see.
>
> Thanks for pointing this out.  We will increase all font sizes.

---

> > ### Comment · Reviewer_jBxN · 2023-08-15
> >
> > Thank you for your reply.
> >
> > I have read the rebuttal by the authors, and I will keep the current rating.

---

### Official Review · Reviewer_gT4m · 2023-07-08

**Soundness:** 3 good
**Presentation:** 1 poor
**Contribution:** 3 good
**Rating:** 4
**Confidence:** 3

**Summary:**

This paper proposes a new construction of time-uniform confidence bands for CDF, where the standard toolkit such as Glivenko—Cantelli cannot be applied due to nonstationarity.
At a high level, the proposed method combines confidence sequence for a certain subset of values via a union bound.
Despite the impossibility result from the adversarial online learning scenario, the proposed construction of confidence bands is shown to have vanishing width for a class of smooth distributions (Theorem 3.3).
This result holds for unbounded random variables (Section 3.3) as well as under a known distribution shift case (Section 3.4).
The simulations demonstrate that the proposed algorithms can be effective.


**Strengths:**

The considered problem of constructing confidence bands for CDFs under nonstationarity is of great practical importance.
The proposed technique seems to be new and interesting. The theoretical guarantee in Theorem 3.3 is insightful as it characterizes the adaptivity.


**Weaknesses:**

- While I believe that the proposed method is new and novel, the writing is hard to follow, which can be much improved.
- The paper is not very self-contained. For example, Table 1 needs to be elaborated further. For example, there is no description on “$w_{\max}$-free” in the main text or in the caption.
- The figures can be revised further. Legends and labels are too small and not very visible. And it might be better to put Figures 2-5 after Section 4 as these are never referred before the experimental section.


**Questions:**

Most of my questions are around Algorithm 1, which seems to contain the key idea of the proposed method. The authors should revise the manuscript so that the key ideas become clear.
- Is i.i.d.ness is assumed throughout Section 3? It is never explicit except in line 80 before Section 3.4.
- In Algorithm 1, aren’t the functions $\epsilon(d)$ and $\Xi_t$ also “inputs” (or “hyperparameters”) to the algorithm? If so, I think it’d better to put it in the algorithm, not in the caption. Further, I think that making Algorithm 1 illustrative by giving a concrete example of $\epsilon$ and $\Xi_t$ may improve readability. Since the paper is extremely notation-heavy, it is not very clear how to parse the algorithm at that abstract level.
- In Algorithm 1, most of the readers will be confused with $W_{1:t}$, as it hasn’t been defined by then. I can search over the text and find it in Section 3.4 that it means the importance weight, but it only distracts a reader if it’s put in Algorithm 1.
- In the caption of Figure 1, it is stated that “The algorithm searches over all d to optimize the overall bound via a provably correct early termination criterion”. What do you mean by the “provably correct early termination criterion”? In Algorithm 1, where is this corresponded? And where do you provide a provably correctness?

As the resulting estimator seems new and interesting, I am inclined to vote for accept, but only provided that the writing is greatly improved in the revision.


**Limitations:**

Limitations are not discussed.

---

> ### Author Rebuttal · Authors · 2023-08-07
>
> > The paper is not very self-contained. For example, Table 1 needs to be elaborated further. For example, there is no description on &ldquo;$w_{\max}$-free&rdquo; in the main text or in the caption.
>
> We will expand section 5 to explicitly define the six properties we identify in the columns of Table 1, including what "unbounded importance weights" refers to.
>
> > The figures can be revised further. Legends and labels are too small and not very visible.
>
> Thanks for pointing this out.  We will increase the sizes of all fonts.
>
> > Is i.i.d.ness is assumed throughout Section 3? It is never explicit except in line 80 before Section 3.4.
>
> The goal of our paper is to dispense with the i.i.d. assumption, so we do not assume it for any of our results. We will be more explicit about this in the introduction. The goal of line 80 was only to point out that a previous proof method used in the i.i.d. setting appears to break down, which is why we use a different technique.
>
> > If so, I think it’d better to put it in the algorithm, not in the caption.
>
> Good suggestion, we'll make this change.
>
> > In Algorithm 1, most of the readers will be confused with, as it hasn’t been defined by then.
>
> We will remove it.
>
> > In the caption of Figure 1, it is stated that “The algorithm searches over all d to optimize the overall bound via a provably correct early termination criterion”. What do you mean by the “provably correct early termination criterion”? In Algorithm 1, where is this corresponded? And where do you provide a provably correctness?
>
> Thanks for pointing this out.  We will include an explicit proof.  The proof sketch is as follows: once the termination criterion is met, any subsequent (deeper in the tree) bounds are evaluated on the same statistics but with a worse confidence level, and hence are dominated.

---

> > ### Comment · Reviewer_gT4m · 2023-08-16
> >
> > I appreciate the authors' response and read other reviews. Though I believe that the paper has an interesting contribution, I think the manuscript needs a major rewriting which will require another round of review. Hence I will keep the score, and I hope the reviewer to revise the whole manuscript carefully so that the problem setting, assumptions, and the description of the algorithm clearer to make the manuscript well-received to a broader audience.

---

### Official Review · Reviewer_hk3u · 2023-07-25

**Soundness:** 2 fair
**Presentation:** 2 fair
**Contribution:** 2 fair
**Rating:** 4
**Confidence:** 2

**Summary:**

The paper presents the time and value uniform bounds on the CDF of the running averaged conditional distribution of a real valued random variable. The new bounds do not require iid setting and always achieve non-asymptotic coverage. The converge speed depends on the smoothness of distribution against the reference distribution, which it is the uniform distribution. Following Howard et al (2021), a confidence sequence of a discrete time random process was employed, and a time uniform coverage property was a key to construct the uniform bounds.
Authors established the bounds for unit interval random variables, extended to real random variables then extended to importance weighted random variables. Algorithms for lower/upper bounds and theoretical properties are presented and simulation studies are provided.


**Strengths:**

Reliable bounds on the CDF for dependent data setting is valuable as iid setting does not always hold. In order to deal with dependent data, the smoothness of distribution to a reference measure is assumed and the time-value uniform bounds are presented. It is examined numerically for both iid samples and nonstationary data.

**Weaknesses:**

This is not my research area and, I found the paper was hard to read and follow. I have an impression that this paper is written assuming that readers have some knowledge and, detailed explanation would be helpful.


i) Confidence sequences Lambda and Xi have very important roles but I can’t grab what they are from the Algorithms 1 and 2 and the justification. Although the confidence sequence review is given in Appendix A, it is not straight forward to make connections to Alg 1 and the rest of paper.

ii) Consequently, it was difficult to follow proofs without fully understanding the confidence sequences and their theoretical properties. Where are the Eqn 12 from? Could authors derive this from the assumptions and properties of the confidence sequence?

iii) Authors did extensive simulation studies with iid samples although the highlight of result is dropping iid assumption.

iv) Some abbreviations were not defined. For example, NSM, DKW and DDRM.

v) Careful edit seems to be needed. For example, there are Eqn (10)s, S_t is not defined in Appendix A.



**Questions:**

Please see the weakness section.

**Limitations:**

Authors addressed the limitation of the new bounds.

---

> ### Author Rebuttal · Authors · 2023-08-07
>
> > This is not my research area and, I found the paper was hard to read and follow.
>
> Thanks for the honest feedback.  We are trying to promote confidence sequences within the machine learning community (they are better known within the statistics literature) and it is challenging to properly calibrate the exposition, but we feel the effort is worthwhile.
>
> > i) Confidence sequences Lambda and Xi have very important roles but I can’t grab what they are from the Algorithms 1 and 2 and the justification. Although the confidence sequence review is given in Appendix A, it is not straight forward to make connections to Alg 1 and the rest of paper.
>
> We will expand Appendix A to better introduce confidence sequences. We will state the definition of a fixed-sample confidence interval and then make the extension to a confidence sequence as a sequence of confidence intervals satisfying a coverage guarantee which holds uniformly over time. We hope this will make the paper more self-contained.
>
> > iv) Some abbreviations were not defined. For example, NSM, DKW and DDRM.
>
> Thanks for pointing this out.
> * We'll drop the usage of the acronym NSM entirely and use "construction" instead (NSM is a term of art from the stats community which is not helpful here.)
> * DKW is defined on line 55.
> * DDRM is itself the name of a technique; when first mentioned in the text a reference is provided, but the potential first encounter for the reader is in the caption of a figure 5, so we'll put a reference on that usage.
>
> > v) Careful edit seems to be needed. For example, there are Eqn (10)s, S_t is not defined in Appendix A.
>
> Thanks for pointing that out, when we moved some material to the appendix we lost some definitions, which we will restore.

---

> > ### Comment · Reviewer_hk3u · 2023-08-16
> >
> > I appreciate the authors' response to my comments and other reviewer's comments. The paper has potential and will be in a better shape after accommodating comments. However, I think it needs a major revision and I will keep the score.

---

> > > ### Author Response · Authors · 2023-08-16
> > > **Actionable ideas?**
> > >
> > > > However, I think it needs a major revision and I will keep the score.
> > >
> > > Thanks for your honest feedback.  It doesn't sound like you object on a technical level, but rather to the presentation.  Do you have some specific suggestions on how we can better describe the concepts to this audience given the page limit?

---

### Author Rebuttal · Authors · 2023-08-07

We thank for reviewers for helping to improve the paper.

To improve the intelligibility of Algorithm 1, we propose adding the attached figure. Further, in the appendix, we will manually describe the execution of Algorithm 1 step-by-step until termination on a simple dataset of five items.

---

### Decision · Program_Chairs · 2023-09-21

**Decision:**

Accept (poster)

**Comment:**

This paper is a a non-trivial extension of the Howard and Ramdas 2022 paper, to time-varying processes. This is a useful extension from the applicability perspective, and the proof techniques differ in a few important ways in order to handle non-stationarity. The most critical feedback from reviewers is on presentation. While the material may be accessible to those who have a background in the relevant literature, the authors are advised to take more care in explaining their notation, particularly by elaborating the time-averaged nature of the quantities and by making sure each concept used is defined ahead of time. The authors are also advised to provide brief proof sketches that highlight the methodology employed, to give a roadmap to the detailed proofs in the supplemental material. Please also elaborate the introduction as far as the relationship with other works goes, e.g., by making the comparisons and contrasts of Table 1 more explicit in the text and including the differences in tools used to achieve the results. The technical contributions of this work do warrant being shared with the community, however it is critical that all the clarity concerns of the reviewers and their other comments are taken into account.